# MAGNet: Motif-Agnostic Generation of Molecules from Scaffolds

**Leon Hetzel**[*1-3]**, Johanna Sommer**[*1,2]**, Bastian Rieck**[1,3,4]**, Fabian Theis**[1-3] **& Stephan Günnemann**[1,2]

[1] School of Computation, Information and Technology, Technical University of Munich
[2] Munich Data Science Institute, Technical University of Munich
[3] Center for Computation Health, Helmholtz Munich
[4] Department of Computer Science, University of Fribourg
{l.hetzel, jm.sommer, b.rieck, f.theis, s.guennemann}@tum.de

## Abstract

Recent advances in machine learning for molecules exhibit great potential for facilitating drug discovery from *in silico* predictions. Most models for molecule generation rely on the decomposition of molecules into frequently occurring substructures (motifs), from which they generate novel compounds. While motif representations greatly aid in learning molecular distributions, such methods fail to represent substructures beyond their known motif set, posing a fundamental limitation for discovering novel compounds. To address this limitation and enhance structural expressivity, we propose to separate structure from features by abstracting motifs to scaffolds and, subsequently, allocating atom and bond types. To this end, we introduce a novel factorisation of the molecules' data distribution that considers the entire molecular context and facilitates learning adequate assignments of atoms and bonds to scaffolds. Complementary to this, we propose MAGNet, the first model to freely learn motifs. Importantly, we demonstrate that MAGNet's improved expressivity leads to molecules with more structural diversity and, at the same time, diverse atom and bond assignments.

## 1 Introduction

Generative models have become a powerful tool for generating novel compounds, finding applications in fields like drug discovery, material science, or chemistry (Bian & Xie, 2021; Butler et al., 2018; Choudhary et al., 2022; Hetzel et al., 2022; Moret et al., 2023). Such deep learning-based molecular generators offer a promising avenue for efficiently navigating the chemical space and generating unique molecules with specific properties (Zhou et al., 2019; Hoffman et al., 2022). A crucial factor contributing to the impressive performance of these models is the incorporation of molecular fragments, known as *motifs*. Motifs make it possible to explicitly include complex structures, such as cycles, into a model and provide a powerful inductive bias for the generative process (Sommer et al., 2023; Wollschläger et al., 2024; Jin et al., 2018; Maziarz et al., 2022; Geng et al., 2023; Kong et al., 2022). However, motif-based approaches often lead to a prohibitively large vocabulary if all motifs are included. As a result, these methods resort to including only the top-$k$ most frequently occurring motifs (Maziarz et al., 2022; Kong et al., 2022), claiming that remaining complex structures can be decoded atom-for-atom.

This claim, however, does not hold in practice, and we demonstrate that a vocabulary of the most commonly occurring motifs is insufficient to capture the vast structural diversity of the molecular space. A simple experiment exposes this shortcoming: As demonstrated in Fig. 1 a, state-of-the-art molecular models fail to reconstruct key structures of FDA-approved drugs. This is because these structures are absent from its vocabulary. If a model struggles to handle structures of known drugs that cannot be trivially built from its vocabulary, how can it be effective in discovering new drugs? This inability raises significant questions about the usefulness of previous generative models and calls for a reassessment of molecular generators from the perspective of structural diversity.

---

[*]Equal contribution
 Project Page: www.cs.cit.tum.de/daml/magnet

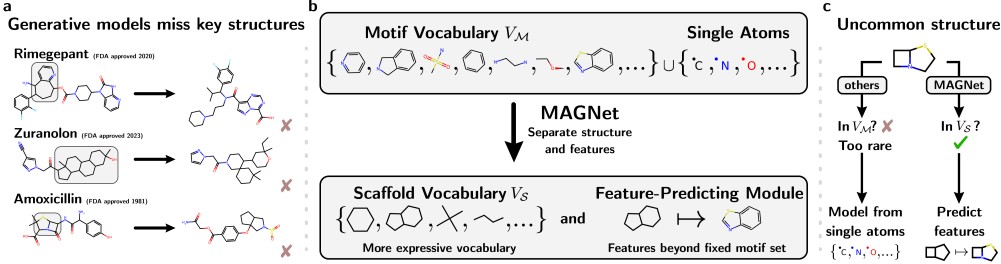

Figure 1: (**a**) Motif-based generative models fail to capture key molecular structures, which limits their expressivity. (**b**) MAGNet overcomes these limitations by separating structures and features. The reduced inductive bias allows for a more expressive vocabulary while enabling the modeling of a wider range of molecular structures by freely learning the mapping from scaffolds to motifs. (**c**) This separation is particularly relevant for "uncommon structures" that pose difficulties for state-of-the-art models; see App. D.2.

We address this fundamental limitation of previous motif-based methods by separating the molecular structure from its features. We achieve this by abstracting motifs to their more general representation: *scaffolds*. In our case, these scaffolds are related to generic Murcko scaffolds but technically do not hold any feature information about bonds or atoms, see Fig. 1 b. Hence, our approach reduces the combinatorial complexity needed to effectively capture the molecules' structural diversity, thereby enhancing expressivity within a fixed-size vocabulary. At the same time, it requires learning the features, *i.e.* atom and bond types, enabling the generation of motifs beyond those contained in a fixed set.

To realise a generative model based on this abstraction, we derive a novel factorisation of the data distribution that splits a molecule into distinct components. The resulting factorisation is utilised in two ways. First, we develop a new fragmentation scheme to generate a structurally expressive scaffold vocabulary. Second, we introduce a new hierarchical method, MAGNet, designed to facilitate the sampling of diverse molecular structures and enable the free learning of scaffold features into motifs, thereby generating molecules in a *motif-agnostic* manner.

Furthermore, we find that established evaluation benchmarks do not sufficiently evaluate the structural diversity of generated molecules. To address this gap, we propose several new analyses to complement existing distribution learning benchmarks. Our primary focus lies on evaluating a model's ability to reliably decode uncommon molecular structures, such as those illustrated in Fig. 1 c, including also those rare substructures that were encountered during training. Additionally, we assess models at the motif level, comparing MAGNet's feature prediction to its motif-based competitors. We specifically examine how diverse and accurate the generated allocations of atom and bond types are.

In summary, our contributions are:

- We improve structural expressivity by abstracting motifs to scaffolds. To this end, we introduce a novel factorisation scheme that facilitates the creation of a more expressive scaffold vocabulary as well as a novel hierarchical generation procedure, MAGNet.
- MAGNet freely learns the featurisation of scaffolds, enabling the generation of a greater variety of atom and bond type allocations than motif-based approaches.
- We propose alternative evaluation analyses for effectively assessing a model's ability to decode complex and uncommon substructures at both the scaffold and motif level.

## 2 RELATED WORK

**Molecule generation** Existing generative models can be divided into three categories (Zhu et al., 2022; Yang et al., 2022; Du et al., 2022): (1) string-based models, relying on string representations like SMILES or SELFIES (Gómez-Bombarelli et al., 2018; Segler et al., 2018; Flam-Shepherd et al., 2022; Fang et al., 2024; Adilov, 2021; Grisoni, 2023), which do not leverage structural information, (2) graph-based models, which model the molecular graphs, and (3) geometry-based models, which

represent molecules by atomic point clouds (Luo & Ji, 2022; Ragoza et al., 2020; Satorras et al., 2021; Luo et al., 2021a; Gebauer et al., 2022; 2019; Hoogeboom et al., 2022; Xu et al., 2023; Huang et al., 2022; Vignac et al., 2023b; Huang et al., 2024; Qiang et al., 2023). Within the area of molecular geometry generation, Adams & Coley (2023), Chen et al. (2023) and Long et al. (2022) focus on the shape-constrained design of conformers within a 3D context and consider surface areas as generation targets. These molecular geometry generators can, in turn, be integrated into frameworks like the one proposed by Ayadi et al. (2025), facilitating their adaptation to various downstream tasks in drug discovery. Moreover, graph-based approaches involve models that represent molecular graphs (i) primarily at the atom level or (ii) predominantly through motifs. Zhu et al. (2022) categorise the generation process further into *sequential* methods, building molecules per fragment while conditioning on a partial molecule (Khemchandani et al., 2020; Shi* et al., 2020; Popova et al., 2019; Mercado et al., 2021; Luo et al., 2021b; Liu et al., 2018; Li et al., 2018; Assouel et al., 2018; You et al., 2019; Yang et al., 2021; Lim et al., 2020; Kajino, 2019; Jin et al., 2020; Bengio et al., 2021; Ahn et al., 2021; Shirzad et al., 2022), and *one-shot* (OS) approaches that create each aspect of the molecular graph in a single step (Kong et al., 2022; Simonovsky & Komodakis, 2018; Ma et al., 2018; Liu et al., 2021; De Cao & Kipf, 2018; Zang & Wang, 2020; Bresson & Laurent, 2019; Flam-Shepherd et al., 2020; Samanta et al., 2020). Note that diffusion-based models iteratively refine the entire graph, making them difficult to categorise as sequential or OS. We refer the reader to App. C for a more detailed discussion on the classification of models into the one-shot or sequential category. While these models are predominantly used in the 3D context, Vignac et al. (2023a) and Ketata et al. (2024) propose diffusion processes that fit category (2).

**Fragmentation and scaffold representation**   Various techniques are available for constructing fragment vocabularies, with a distinction between chemically-inspired and data-driven approaches. For example, both HierVAE (Jin et al., 2020) and MoLeR (Maziarz et al., 2022) adopt a heuristic strategy known as "breaking bridge bonds" to decompose molecules into rings and remainder fragments, emphasising chemically valid substructures. In a similar vein, JT-VAE (Jin et al., 2018) employs fragmentation guided by the construction of junction trees. In contrast, PS-VAE (Kong et al., 2022) and MiCaM (Geng et al., 2023) take a data-driven bottom-up approach, creating fragments by merging smaller components, starting from single atoms. MiCaM even integrates attachment points, resulting in a larger, "connection-aware" vocabulary.

## 3   FACTORISING THE DATA DISTRIBUTION $\mathbb{P}(\mathcal{G})$

We provide a general mathematical description of our framework below. However, this section is not essential for understanding the main concepts of the remainder of this work and we introduce the matching model in Sec. 4. We consider a molecule $\mathcal{G}$ to be defined by its underlying graph, i.e., by its graph structure (nodes and edges), together with its node and edge features, describing atoms and bonds, respectively. We consider a factorisation of the probability mass function $\mathbb{P}(\mathcal{G})$, which is illustrated in Fig. 2, that decouples structure from features:

$$\mathbb{P}(\mathcal{G}) = \mathbb{P}(\mathcal{G} \mid \mathcal{G}_\mathcal{S})\,\mathbb{P}(\mathcal{G}_\mathcal{S}) \quad , \text{ since } \quad \mathbb{P}(\mathcal{G}_\mathcal{S} \mid \mathcal{G}) = 1 \quad ,$$

where $\mathcal{G}$ refers to the full atom-level graph and $\mathcal{G}_\mathcal{S}$ to its abstraction to scaffolds. $\mathcal{G}_\mathcal{S}$ represents a coarse view of a molecule's topology by specifying the scaffolds that make up the molecule as well as their connectivity. Representing $\mathcal{G}_\mathcal{S}$ by the multiset of scaffolds $\mathcal{S}$ and their (typed) connectivity $A \in \mathcal{A}^{|\mathcal{S}| \times |\mathcal{S}|}$, where $\mathcal{A}$ defines the possible values of $A$, $\mathcal{G}_\mathcal{S} = (\mathcal{S}, A)$, $\mathbb{P}(\mathcal{G}_\mathcal{S})$ becomes:

$$\mathbb{P}(\mathcal{G}_\mathcal{S}) = \mathbb{P}(A \mid \mathcal{S})\,\mathbb{P}(\mathcal{S}) \,.$$

Moving forward to the atom level, a node $S$ in the scaffold graph $\mathcal{G}_\mathcal{S}$ can be expanded into its binary adjacency matrix of its $s = |S|$ nodes, i.e. $S \in \{0, 1\}^{s \times s}$ and is equipped with node and edge features, corresponding to atom and bond types, respectively. We consider this feature-equipped representation of $S$ to be a typed subgraph $M$, or motif, of the input graph $\mathcal{G}$, and denote the associated multiset of motifs within the molecule by $\mathcal{M}$.

The connectivity between two scaffolds, signified by $A_{kl} \neq 0$, indicates that, at the atom level, the two scaffolds share a common join node (atom) $j$. To determine a node $j$, we define the set of join nodes $\mathcal{J}$, which includes all join nodes $j$ that are contained in two motifs: $\mathcal{J} = \{j \mid j \in M_k, \ j \in M_l, \ A_{kl} \neq 0, \ S_k, S_l \in \mathcal{S}\}$.

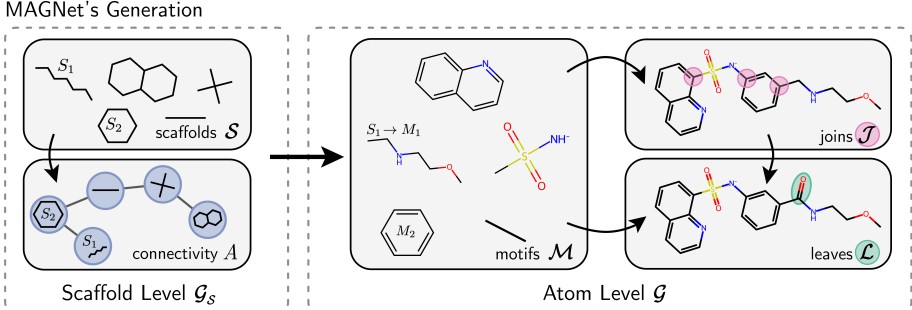

Figure 2: On the scaffold-level, MAGNet predicts the scaffold multiset $\mathcal{S}$ and its connectivity $A$. Progressing to the atom-level, $\mathcal{G}_\mathcal{S}$ informs the generation of motifs $\mathcal{M}$. To fully define the molecular graph $\mathcal{G}$, the join node positions $\mathcal{J}$ and the leaf nodes $\mathcal{L}$ are predicted.

Note that we can effectively provide information about the atom type of the join node $j$ by considering $\mathcal{A} = \{0, C, N, \dots\}$ to include atom types, which is reflected by the ablation study on $\mathcal{A}$ in App. D.7. Lastly, we do not include all atoms in the scaffold graph to allow for a concise set of structurally distinct scaffolds, see Sec. 4.1. Specifically, we define the set of leaf nodes $\mathcal{L}$ to describe nodes $l$ with degree $d_l = 1$ and neighbours $k \in \mathcal{N}_l$ with $d_k = 3$.

In conclusion, $\mathcal{G}$ is defined by the combination of motifs $\mathcal{M}$, join nodes $\mathcal{J}$, and leaf nodes $\mathcal{L}$, enabling to factorise $\mathbb{P}(\mathcal{G})$ as follows:

$$\begin{aligned}
\mathbb{P}(\mathcal{G}) &\coloneqq \mathbb{P}(\mathcal{L}, \mathcal{J}, \mathcal{M}) \\
&= \mathbb{P}(\mathcal{L} \mid \mathcal{M}, \mathcal{J}) \mathbb{P}(\mathcal{J} \mid \mathcal{M}, A) \, \mathbb{P}(\mathcal{M} \mid \mathcal{G}_\mathcal{S}) \, \mathbb{P}(\mathcal{G}_\mathcal{S}) \, ,
\end{aligned}$$

where we use $\mathcal{G}_\mathcal{S}$ to factorise the distribution. Note that $\mathcal{J}$ is *conditionally independent* of $\mathcal{S}$ given the motifs $\mathcal{M}$. Similarly, $\mathcal{L}$ is conditionally independent of $A$ given $\mathcal{J}$.

## 4 MODELLING MOLECULES FROM SCAFFOLDS

Building upon the presented factorisation, we continue to introduce our method to derive a concise scaffold vocabulary and discuss the matching generative model: MAGNet. MAGNet is a graph-based model that employs a unique approach by generating each hierarchy level in a single step. It positions itself between the traditional categories of single-atom and fragment-based models by utilising scaffolds as building blocks and subsequently generating appropriate atom and bond attributes. To facilitate this generation of diverse motifs and molecular structures, our fragmentation is designed to achieve concise representations.

### 4.1 IDENTIFYING A CONCISE SET OF SCAFFOLDS $\mathcal{V}_S$ FROM DATA

Given a dataset of molecules, our fragmentation scheme aims to represent a molecule through clear structural elements, for which we provide examples in App. A. For this, we start by removing all leaf nodes $\mathcal{L}$ across the graph, following the approach outlined in previous works (Jin et al., 2020; Maziarz et al., 2022). This step helps to divide the molecule $\mathcal{G}$ into cyclic and acyclic parts. Importantly, instead of modelling the connection between two fragments $M_i$ and $M_j$ with a connecting bond, we represent it by a shared atom, matching the definition of a join node $\nu \in \mathcal{J}$ from Sec. 3.

In a subsequent step, we decompose the resulting acyclic fragments to reduce the number of required scaffolds as much as possible. To this end, we introduce "junctions", acyclic structures that are defined by a center node of degree three or four and its neighbours. When compared to the "Breaking Bridge Bonds" decomposition (Jin et al., 2020; Maziarz et al., 2022), our approach reduces complexity and results in a much smaller vocabulary by collapsing acyclic structures into distinct scaffolds. Additionally, in contrast to data-driven methods like those outlined in Kong et al. (2022) and Geng et al. (2023), our decomposition method maintains structural integrity through its top-down approach.

To arrive at our scaffold vocabulary, we remove atom and bond types from the resulting fragments in the final step. As a suitable generative model has to map a single scaffold to its various representations, this fragmentation further enables us to model smoother transitions between scaffold representations. This is in contrast to fragment-based methods, which have to select different tokens from a large vocabulary when two motifs differ in, e.g. just bond type, see Fig. 6 b in App. A.

## 4.2 MAGNET'S ENCODER

MAGNet is trained as a VAE model (Kingma & Welling, 2022), where the latent vectors $z$ are trained to encode meaningful semantics about the input data, which subsequently inform the generation process. During training, our model is optimised by maximising the ELBO:

$$L = \mathbb{E}_{z \sim \mathbb{Q}}\big[\mathbb{P}(\mathcal{G} \mid z)\big] + \beta D_{\text{KL}}\big(\mathbb{Q}(z \mid \mathcal{G})\big\| P\big)$$
$$\text{with} \quad P \sim \mathcal{N}(0, \mathbb{1})$$

where the KL-divergence $D_{\text{KL}}$ serves to regularise the posterior $\mathbb{Q}(z \mid \mathcal{G})$ towards similarity with the Normal prior $P$ in the latent space, weighted by $\beta$.

MAGNet's encoder aims to learn the approximate posterior $\mathbb{Q}(z \mid \mathcal{G})$. At its core, the encoder leverages a graph transformer (Shi et al., 2021) for generating node embeddings of the molecular graph. Since MAGNet generates molecules in a coarse to fine-grained fashion, we encode information about the decomposition of the molecules. We compute individual embeddings for the molecular graph $\mathcal{G}$ and the scaffold graph $\mathcal{G}_\mathcal{S}$ by aggregating over the corresponding atom and scaffold nodes, respectively. In addition, we separately aggregate all join and leaf nodes. The resulting representation of these individual components—molecular graph, scaffolds, join nodes, and leaf nodes—is then mapped to the latent space, constituting the graph embedding $z_\mathcal{G}$. More details about the chosen node features as well as technical specifications of the encoder can be found in App. B.1.

## 4.3 MAGNET'S GENERATION PROCESS

MAGNet is designed to represent the hierarchy of the factorisation into scaffold- and atom-level from Sec. 3, see Fig. 2. That is, given a vector $z$ from the latent space, MAGNet's generation process first works on the scaffold-level to predict $\mathcal{G}_\mathcal{S}$, defined by the multiset $\mathcal{S}$ and its connectivity $A$, before going to the atom level defined by the motifs $\mathcal{M}$, join nodes $\mathcal{J}$, and leaf nodes $\mathcal{L}$. Additional details on the implementation can be found in App. B.2.

**Scaffold-Level** On the scaffold-level, MAGNet first generates the **scaffold multiset** $\mathcal{S}$ via a transformer decoder module—the same scaffold can occur multiple times in one molecule—from the latent representation $z$. More specifically, we learn $\mathbb{P}(\mathcal{S} \mid z)$ by conditioning the generation on the latent code $z$ and generate one scaffold at a time, conditioning also on the intermediate representation of the scaffolds.

Given the scaffold multiset $\mathcal{S}$, MAGNet infers the **scaffold connectivity** $A$ between scaffolds $S_i, S_j \in \mathcal{S}$ from scaffold set embeddings obtained with a multi-attention module. Formally, we learn $\mathbb{P}(A \mid \mathcal{S}, z) = \prod_{i,j=1}^{n} \mathbb{P}(A_{ij} = t \mid S, z)$ where $t \in \{0, \text{C}, \text{N}, \dots\}$ not only encodes the existence (or absence) of a scaffold connection but also its atom type. We consider a typed version of $A$ to provide a meaningful condition for generating the motifs $\mathcal{M}$ later on and provide and ablation on this in App. D.7. The scaffold-level loss is computed by $L_{\mathcal{G}_\mathcal{S}} = L_\mathcal{S} + L_A$, where $L_\mathcal{S}$ and $L_A$ refer to the categorical losses of the scaffold set and connectivity, respectively.

**Atom-Level** Leveraging a molecule's scaffold-level representation $(\mathcal{S}, A)$, MAGNet incorporates a feature-predicting module to identify appropriate node and edge attributes for each scaffold, see Fig. 1 b. By freely predicting the **atom and bond types** to define the motifs $\mathcal{M}$, MAGNet gains flexibility compared to its motif-based competitors. The mapping of a scaffold $S_i$ to a motif $\mathbb{P}(M_i \mid \mathcal{S}, A, z)$ is performed in two steps, starting with the prediction of the respective atom types $M_i^a$ from the scaffold graph and latent code $z$ by means of a transformer decoder module. Subsequently, the resulting atom embeddings $M_i^a$ are leveraged to determine the matching **bond types** $M^b$ between connected nodes.

By conditioning on $A$, we can ensure that $M_k$ includes all nodes needed for connectivity on the atom-level. This comes from the fact that the motifs $M$ adhere to the join node types defined by $A$. In mathematical terms, this means $A_{kl} \in \bigcup_j M_k^a \cap M_l^a$, where $a$ signifies the exclusive consideration of atoms.

To establish the connectivity on the atom level, MAGNet proceeds to identify the **join nodes** $\mathcal{J}$, whose types are already defined, via a multi-layer perceptron. Therefore, MAGNet only needs to predict the pairs of node positions $p_a$ that form joins, collectively creating the set of join nodes $\mathcal{J}$. Mathematically, we express the likelihood of merging nodes $i$ and $j$ in motifs $M_k$ and $M_l$ by the merge probability $J_{ij}^{(k,l)} = \mathbb{P}(p_i \equiv p_j \mid \mathcal{M}, A, z)$, which constitutes the join matrix $J^{(k,l)} \in [0,1]^{V_{S_k} \times V_{S_l}}$. In the final step, MAGNet utilises a transformer decoder to predict the **leaf nodes** $\mathcal{L}$, which consists of determining the atom type and its attachment to the current atom graph $\mathcal{C}$ (for core molecule). Optimising the likelihood $\mathbb{P}(\mathcal{L}_S \mid \mathcal{C}, z)$ is done similarly to the motif prediction, only that we use the atom graph $\mathcal{C}$ as model input. The final atom-level loss consists of $L_{\mathcal{G}} = L_{\mathcal{M}} + L_{\mathcal{J}} + L_{\mathcal{L}}$, where $L_{\mathcal{M}}$, $L_{\mathcal{J}}$, and $L_{\mathcal{L}}$ describe categorical losses for motifs $\mathcal{M}$, including both atom and bond types, join nodes $\mathcal{J}$, and leaf nodes $\mathcal{L}$, respectively. When optimising MAGNet with a reconstruction loss consisting of $L_{\mathcal{G}} + L_{\mathcal{G}_S}$, we have observed that a simple KL-regularisation alone is inadequate for achieving a smoothly structured latent space. Our analysis in App. B.3 shows that the latent space suffers from over-pruning behaviour (Yeung et al., 2017). To remedy this, we apply a normalising flow post-hoc to the latent space, aligning it more effectively with the prior. To this end, we rely on Conditional Flow Matching (Lipman et al., 2023) and, more specifically, use the version based on minibatch optimal transport as presented by Tong et al. (2024). We specify MAGNet's hyperparameter configuration in App. B.4.

**Limitations**    As for the limitations, we note that both sequential and one-shot molecular generation methods are susceptible to error propagation (Muenkler et al., 2023). Our results in Sec. 5, however, indicate that the separation between scaffold and atom-level does not harm MAGNet in practice, as MAGNet reliably identifies suitable atom and bond allocations for the generated scaffold graph. While MAGNet successfully generates diverse molecular structures, incorporating synthesizability considerations would further enhance its practical applicability in drug discovery pipelines. This limitation could be addressed by training on multi-component reaction datasets, potentially bridging the gap between computational predictions and experimental validation (Graziano et al., 2023).

## 5    RESULTS

We compare MAGNet to state-of-the-art molecular generators across several dimensions of the generation process. In Sec. 5.1, we investigate the reconstruction and sampling of scaffolds $\mathcal{S}$, as the fundamental component of MAGNet's factorisation, and show that our model, unlike the baselines, captures the diverse structural characteristics found in molecules. In Sec. 5.2, we continue to evaluate the generative performance using established benchmarks. In Sec. 5.3, we analyse MAGNet's ability to determine atom and bond allocations $\mathcal{M}$ freely and demonstrate that MAGNet learns to generate a larger variety of motifs in accordance with the dataset compared to motif-based approaches. Finally, in Sec. 5.4, we give an outlook on how MAGNet can be utilised for downstream applications, such as goal-directed generation and conditioning on various levels of the generation process.

**Baselines**    In this section, our focus lies on variational autoencoder models designed to generate molecular graphs. We select models from the one-shot and sequential categories as detailed in Sec. 2. These include PS-VAE (Kong et al., 2022), a two-step generation framework that first generates subgraphs before predicting connections, from the one-shot category and MoLeR, a method that extends a partially generated graph by adding new fragments at each step, (Maziarz et al., 2022) from the sequential category. Additionally, we incorporate MiCaM (Geng et al., 2023) as a third baseline, which is another fragment-based method that utilises connection-aware motifs, leading to a significantly larger and more fine-grained vocabulary than ours. In order to contextualise MAGNet with the full spectrum of molecule generation methods, we also evaluate JTVAE (Jin et al., 2018), HierVAE (Jin et al., 2020), GraphAF (Shi* et al., 2020), SMILES-LSTM (SM-LSTM) (Segler et al., 2018) and CharVAE (Gómez-Bombarelli et al., 2018) as baselines for distribution learning benchmarks.

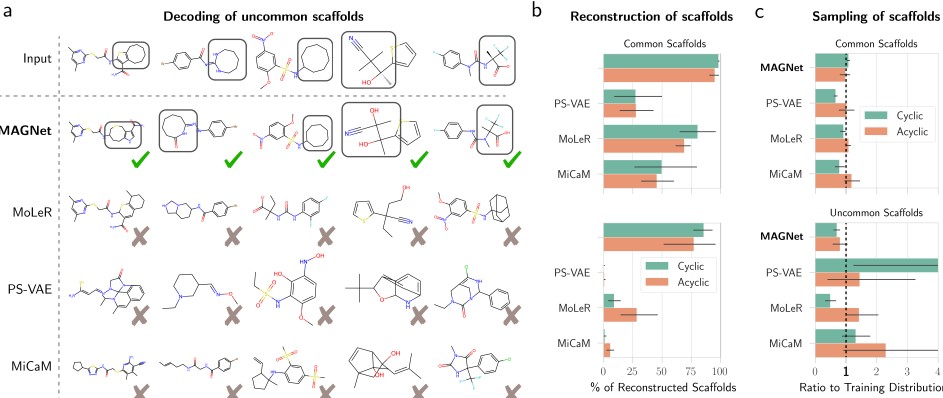

Figure 3: **(a)** Reconstruction of molecules that include large cycles or complex junctions. Relying on individual atoms to build these structures is not sufficient. Only MAGNet is able to reliably decode its latent code $z$. **(b)** Percentage of reconstructed scaffolds. MAGNet substantially improves in reconstructing both common and—more importantly—uncommon scaffolds. **(c)** Comparison of sampled scaffolds to their frequency in the training distribution. A ratio of 1 is optimal.

**Benchmarks and datasets**    To evaluate the ability to learn the underlying distribution of molecules, we employ two standard benchmarks for *de novo* molecule generation. The GuacaMol benchmark asseses the ability of a generative model to sample by the distribution of a molecular dataset (Brown et al., 2019). Next to evaluating the uniqueness and novelty of sampled molecules, the benchmark also computes distributional distances to the reference, i.e. the KL-divergence and Fréchet distance score (FCD) (Brown et al., 2019). We use the MOSES benchmark (Polykovskiy et al., 2020) to report measures for the internal diversity (IntDiv) of generated molecules as well as chemical properties such as synthetic accessibility (SA), the octanol-water partition coefficient (logP), and the viability for drugs (QED). We also evaluate a subset of our baselines on the GuacaMol goal-directed benchmark, which tests the ability of generative models to generate molecules that maximise certain score functions, e.g. similarity to a known drug-like molecule. All models are trained on the ZINC dataset (Irwin et al., 2020) and the benchmarks conducted on the corresponding test set. We further use QM9 (Wu et al., 2018), GuacaMol (Brown et al., 2019), CheMBL (Mendez et al., 2019), and L1000 (Subramanian et al., 2017) for additional evaluations.

## 5.1 USING SCAFFOLDS TO REPRESENT STRUCTURAL VARIETY OF MOLECULES

**Reconstructing complex structures**    As a continuation of Fig. 1, our first experiment provides qualitative insights into how accurately scaffolds are decoded, $\mathbb{P}(\mathcal{S} \mid z)$. We assess the decoder's performance in reconstructing molecules from the test set, which includes uncommon scaffolds like large rings or complex junctions. Our observations reveal that the baseline models have difficulty in constructing complex scaffolds, as illustrated in Fig. 3a. We attribute this limitation to the absence of motifs which map to such scaffolds in their top-$k$ vocabularies. Consequently, these models face the challenge of constructing scaffolds such as large rings from individual atoms. In contrast, our proposed model, MAGNet, operates with a moderately-sized vocabulary that includes complex scaffolds, enabling it to generate molecules that follow the latent code and the corresponding ground truth molecules (see the latent displacement analysis in App. D.4).

**MAGNet reliably decodes scaffolds**    Building on our analysis of large cycles and uncommon junctions, we extend our investigation to assess how effectively different models can reconstruct the scaffold set $\mathcal{S}$ in a general context. Given our focus on $\mathbb{P}(\mathcal{S} \mid z)$, we can disregard the connectivity $A$ and motifs $\mathcal{M}$. As illustrated in Fig. 3b, our findings demonstrate that MAGNet consistently outperforms all evaluated baselines, supporting the hypothesis that the other methods fail to faithfully decode complex molecular structures from single atoms and rely primarily on the information encoded in their vocabulary.

Table 1: GuacaMol and MOSES Benchmark. We report mean and standard deviation using 5 random seeds. We underline the best graph-based method and make the best method within each category **bold**.

| | | GuacaMol | | MOSES | | | |
|---|---|---|---|---|---|---|---|
| | | FCD ($\uparrow$) | KL ($\uparrow$) | IntDiv ($\uparrow$) | logP ($\downarrow$) | SA ($\downarrow$) | QED ($\downarrow$) |
| SM. | CharVAE | $0.17 \pm 0.08$ | $0.78 \pm 0.04$ | $\mathbf{0.88} \pm \mathbf{0.01}$ | $0.87 \pm 0.14$ | $0.48 \pm 0.13$ | $0.06 \pm 0.03$ |
| | SM.-LSTM | $\mathbf{0.93} \pm \mathbf{0.00}$ | $\mathbf{1.00} \pm \mathbf{0.00}$ | $0.87 \pm 0.00$ | $\mathbf{0.12} \pm \mathbf{0.01}$ | $\mathbf{0.04} \pm \mathbf{0.02}$ | $\mathbf{0.00} \pm \mathbf{0.00}$ |
| Sequential | GraphAF | $0.05 \pm 0.00$ | $0.67 \pm 0.01$ | $\underline{0.93} \pm \underline{0.00}$ | $0.41 \pm 0.02$ | $0.88 \pm 0.10$ | $0.22 \pm 0.01$ |
| | HierVAE | $0.53 \pm 0.14$ | $0.92 \pm 0.01$ | $0.87 \pm 0.01$ | $0.36 \pm 0.17$ | $0.20 \pm 0.14$ | $0.03 \pm 0.00$ |
| | MiCaM | $0.63 \pm 0.02$ | $0.94 \pm 0.00$ | $0.87 \pm 0.00$ | $0.20 \pm 0.05$ | $0.51 \pm 0.03$ | $0.08 \pm 0.00$ |
| | JTVAE | $0.75 \pm 0.00$ | $0.94 \pm 0.00$ | $0.86 \pm 0.00$ | $0.28 \pm 0.03$ | $0.34 \pm 0.01$ | $\underline{\mathbf{0.01}} \pm \underline{\mathbf{0.00}}$ |
| | MoLeR | $\underline{\mathbf{0.80}} \pm \underline{\mathbf{0.01}}$ | $\underline{\mathbf{0.98}} \pm \underline{\mathbf{0.00}}$ | $0.87 \pm 0.00$ | $\underline{\mathbf{0.13}} \pm \underline{\mathbf{0.02}}$ | $\underline{\mathbf{0.06}} \pm \underline{\mathbf{0.01}}$ | $\underline{\mathbf{0.01}} \pm \underline{\mathbf{0.01}}$ |
| One-Shot | PSVAE | $0.28 \pm 0.01$ | $0.83 \pm 0.00$ | $\mathbf{0.89} \pm \mathbf{0.00}$ | $0.34 \pm 0.02$ | $1.18 \pm 0.05$ | $0.05 \pm 0.00$ |
| | DiGress | $0.65 \pm 0.00$ | $0.91 \pm 0.00$ | $0.86 \pm 0.00$ | $0.61 \pm 0.02$ | $\mathbf{0.09} \pm \mathbf{0.01}$ | $0.05 \pm 0.01$ |
| | MAGNet | $\underline{\mathbf{0.76}} \pm \underline{\mathbf{0.00}}$ | $\underline{\mathbf{0.95}} \pm \underline{\mathbf{0.00}}$ | $0.88 \pm 0.00$ | $\mathbf{0.22} \pm \mathbf{0.01}$ | $0.12 \pm 0.01$ | $\underline{\mathbf{0.01}} \pm \underline{\mathbf{0.00}}$ |

Even with a significantly increased vocabulary for methods such as MoLeR, we find in App. D.1 that a larger vocabulary does not significantly help to model to reconstruct uncommon scaffolds. Furthermore, we observe that MiCaM, MoLeR, and PS-VAE hallucinate structures dissimilar to the scaffolds present in the dataset, see App. D.8.

**MAGNet matches the scaffold distribution more accurately**    To further analyse to what extent uncommon scaffolds are not only reconstructed but purposefully sampled, we analyse the scaffold set of generated molecules. If the other models are able to represent scaffolds that are not included in their vocabulary, they should be able to reflect the reference distribution of scaffolds. For this evaluation, we decompose sampled molecules into their scaffolds. We then measure the models' over- and undersampling behaviour based on the ratio $r_{S_i} = \frac{c_s(S_i)}{\sum_k c_s(S_k)} \times \frac{\sum_k c_t(S_k)}{c_t(S_i)}$, where $c_t$ and $c_s$ refer to the count function applied to the training and sampled sets, respectively. Fig. 3c shows that baseline methods fail to generate molecules in accordance with the reference scaffold distribution in practice. On common scaffolds, *i.e.* those that occur in more than 10% of the molecules, all evaluated models are able to match the ratio of the data distribution. For uncommon scaffolds, however, the baselines fail: while PS-VAE heavily oversamples both ring-like and chain-like scaffolds, MoLeR and MiCaM oversample chain-like scaffolds. MAGNet matches the reference distribution best across categories and we conclude that the abstraction to scaffolds benefits generation.

## 5.2 GENERATIVE PERFORMANCE EVALUATED ON COMMON BENCHMARKS

**MAGNet performs on par with motif-based approaches**    The MOSES and GuacaMol benchmarks are conducted on $10^4$ latent codes sampled from the prior distribution, $z \sim P$, and decoded into valid molecules. Our results for both benchmarks are depicted in Tab. 1, where we classify the methods into their generative approaches as described in Sec. 2. We do not report Novelty and Uniqueness, as almost all evaluated models achieve 100% on these metrics. Solely GraphAF and HierVAE perform slightly worse but still achieve above 90% in Uniqueness and Novelty. For the baselines DiGress, SM-LSTM, and CharVAE, which are not able to achieve 100% Validity, we sample until we obtain $10^4$ valid molecules. We report all benchmark metrics in App. D.3. While MoLeR sets the state of the art on both FCD and KL, MAGNet outperforms all other graph-based baselines. This supports the proposed factorisation in Sec. 3 while also challenging the common perception that methods for molecule generation must rely on motif vocabularies to obtain good generative performance.

**The FCD metric is insufficient for evaluating structural diversity**    Despite the FCD being an important metric for molecular distribution learning, we find that it fails to provide insights about the structural diversity of the generated molecules. Evaluating the benchmark on a subset of $10^4$ molecules from the training data, which was filtered to include only the 10 most common scaffolds, results in an FCD score of $0.89$. This observation offers an explanation for why models like MoLeR can achieve state-of-the-art FCD scores, despite not accurately capturing the scaffold distribution, as demonstrated in Sec. 5.1. This underscores that our analysis on structural diversity complements these benchmarks, providing valuable insights about the molecular distribution.

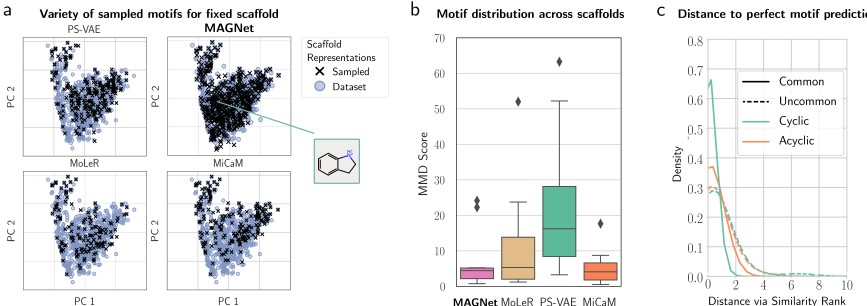

Figure 4: (**a**) Example of generated fragments by MAGNet and baseline methods. (**b**) MMD computation to quantify similarity between generated and ground truth motifs. (**c**) Rank comparison between predicted fragments and their original counterparts.

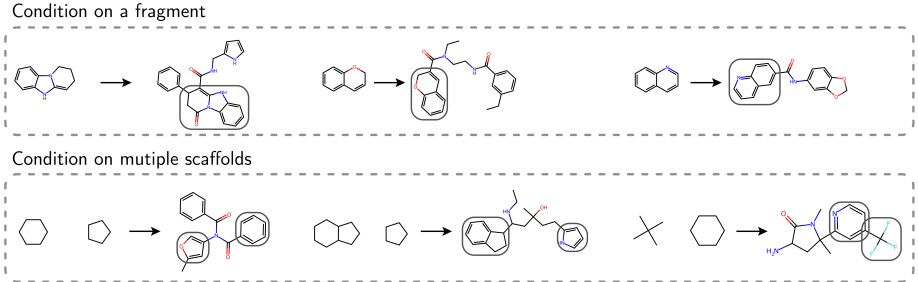

Figure 5: Examples of conditional molecule generation with MAGNet. The generation is conditioned on (*top*) a complete fragment, including atoms and edges, and (*bottom*) two distinct scaffolds.

## 5.3 GENERATION OF MOTIFS $\mathcal{M}$

Having established that MAGNet reliably decodes molecular structures and their scaffolds and samples diversely, we continue to evaluate MAGNet on the motif level.

**MAGNet's generated motifs are superior to fixed fragments**     The larger a given scaffold, the more the combinatorial aspect starts to dominate: with a size-limited vocabulary, it is challenging to reflect the diversity of a scaffold's realisations to motifs during decoding. This is shown in Fig. 4a, which provides a qualitative view on generated and sampled motifs. We extract the motif of a single scaffold from the molecules sampled in Tab. 1 and plot the two principal components of their fingerprints. For the chosen scaffold alone, there are 791 representations in ZINC. PS-VAE, MiCaM, and MoLeR do not cover the distribution fully, even though the scaffold appears commonly in the dataset. MAGNet, however, covers all parts of the distribution, even outliers. Fig. 4b shows the MMD quantification of the procedure shown in Fig. 4a, confirming that MAGNet is able to cover the entire distribution of motifs best. Note that, by learning to predict the bond and atom assignments, MAGNet achieves coverage of motifs that extend upon the fixed motif sets of the other methods. This effect becomes also apparent when assessing the zero-shot generalisation capabilities on other datasets, where MAGNet is able to achieve the highest similarity scores, improving over the strongest baseline by up to 20%, see App. D.6.

**Allocation of atom and bonds to scaffolds**     Extending the sampling analysis from Fig. 4b, we quantify the process of turning a scaffold into a chemically valid motif in Fig. 4c. For each scaffold in the dataset, we compute the similarities between the set of all generated and ground truth motifs. Within the set of ground truth motifs and successful scaffold decodings, we compute the similarity rank between generated and ground truth pairs. In the majority of cases, MAGNet achieves rank 0 or 1 in the motif generation, indicating near-perfect accuracy, with uncommon rings remaining the most challenging to decode.

## 5.4 MAGNET FOR DOWNSTREAM APPLICATIONS

Having analysed the generative performance of MAGNet and the benefit of the proposed scaffold fragmentation, we demonstrate that, like all VAE-based architectures, MAGNet can be adapted to generate molecules with desired properties. Additionally, we outline how one can use MAGNet's factorisation for the generation of linkers and structure-constrained generation.

**Goal-directed generation**    We evaluate to what extent the models can be used to find molecules that maximise a given score function. We compare against MoLeR and MiCaM and additionally provide a random baseline that samples molecules from the dataset as lower bound. Tab. 2 shows the results and we observe that MAGNet performs on par with or better than MoLeR, confirming our competitive performance on established benchmarks. We provide more information about the latent optimisation procedure as well as additional goal-directed results in App. D.9.

Table 2: Results for the GuacaMol goal-directed benchmarks. All scores should be maximised. We report mean and standard deviation and highlight the best method in **bold**.

|  | MAGNet | MoLeR | MiCaM | Random |
|---|---|---|---|---|
| Isomers | **0.34 ± 0.03** | 0.32 ± 0.03 | 0.31 ± 0.04 | 0.17 ± 0.12 |
| MPO | **0.74 ± 0.06** | 0.70 ± 0.08 | 0.71 ± 0.07 | 0.61 ± 0.22 |
| Property | **0.82 ± 0.02** | 0.81 ± 0.03 | **0.82 ± 0.01** | 0.34 ± 0.30 |
| Redisc. | 0.17 ± 0.00 | **0.24 ± 0.00** | **0.24 ± 0.00** | 0.14 ± 0.01 |
| Similarity | **0.43 ± 0.03** | 0.38 ± 0.04 | 0.38 ± 0.03 | 0.26 ± 0.08 |

**MAGNet efficiently generates molecules conditioned on fragments and scaffolds**    In the context of potential downstream applications, we investigate novel scaffold conditioning methods made possible by MAGNet's factorisation. Besides the latent space interpolation in App. D.5, Fig. 5 illustrates that MAGNet is capable to condition not only on a single scaffold but also on multiple scaffolds, even when they are not directly connected within the resulting molecule. MAGNet makes this possible by enforcing e.g. a set of conditioning scaffolds or given scaffold representations during the decoding phase. This poses a significant challenge for models like MoLeR, which rely on extending connected subgraphs for scaffold conditioning. Moreover, MAGNet enables conditioning on multiple levels of abstraction and can generate molecules conditioned on a fragment as well as solely based on a scaffold, see Fig. 5.

## 6 CONCLUSION

To address fundamental limitations of motif-based approaches for molecular graph generation, we propose to separate structure from features. To this end, we introduce a novel graph factorisation that enables us to build an expressive scaffold vocabulary using a novel fragmentation scheme. Furthermore, we present MAGNet, a generative model that builds upon this abstraction to achieve greater structural expressivity. Our experiments demonstrate that MAGNet outperforms existing motif-based models in terms of structural diversity and performs competitively on established benchmarks. We argue that a hierarchical approach, such as the one adopted in MAGNet, is crucial for effectively leveraging scaffold representations. However, modifications to this approach could be promising for future advancements building upon our proposed method.

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

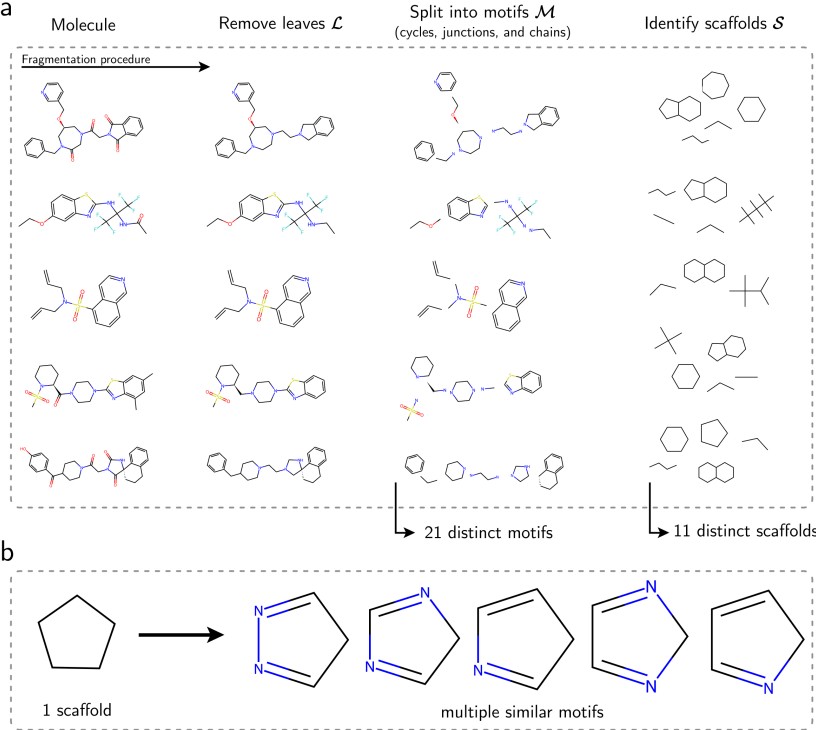

Figure 6: **(a)** Examples of the fragmentation procedure. Starting with the entire molecules, the fragmentation first removes leaf nodes $\mathcal{L}$, continues to split into motifs $\mathcal{M}$, and then identifies distinct scaffolds $\mathcal{S}$. **(b)** Example for a single scaffold that has multiple similar representations in terms of atom and bond types, illustrating how our scaffold abstraction can reduce combinatorial explosion and enable smooth learning.

# A   ABSTRACTION TO SCAFFOLDS AND DETAILS ON PROPOSED FRAGMENTATION

Fig. 6 a shows examples of the proposed fragmentation and abstraction to scaffolds. First, we identify leaf nodes $\mathcal{L}$ and then divide the core molecule $\mathcal{C}$ into structurally distinct fragments $\mathcal{M}$ that can be categorised into rings, chains, and junctions. Note that adjacent scaffolds share a join node $v \in \mathcal{J}$ instead of being connected through a bond. This representation of connectivity between fragments is advantageous compared to the "Breaking Bridge Bonds" decomposition (Jin et al., 2020; Maziarz et al., 2022), as the separation of motifs, such as rings and chains, does not require truncating the chain. Given this fragmentation, atoms can simultaneously be part of a ring and a chain, and MAGNet accounts for that.

Fig. 6 a illustrate that many fragments share the same topology but differ in the atom and bond types. Extracting a scaffold $S_i$ from its motif $M_i$ means to reduce the typed adjacency to its binary connectivity, discarding any node features. After creating a vocabulary using all unique scaffolds from the dataset, we can check their isomorphism by comparing their hashes (Leman & Weisfeiler, 1968).

By this abstraction, MAGNet can learn smoother transitions between different scaffold representations. Fig. 6 b showcases a simple example of this: the shown motifs share very similar sets of atoms and bonds, as well as their underlying structure, but they differ in the exact positions of atoms and bonds. Fragment-based methods would be required to replace the motif token entirely, having to choose its replacement from a potentially large vocabulary. By disentangling structure from features, we enable MAGNet to learn such transitions smoothly.

## B DETAILS MAGNET

### B.1 ENCODER

We build the node features that are processed in MAGNet's encoder from different attributes, see Tab. 3. We include the atom type ('atom_id_dim'), its charge ('atom_charge_dim'), as well as its multiplicity value ('atom_multiplicity_dim'). We proceed accordingly for the scaffold level and include the scaffold id ('scaffold_id_dim'), its multiplicity ('scaffold_multiplicity_dim'), as well chemical features ('motif_feat_dim') computed through RDKit (Landrum & others, 2013). Since the latter are not learned during training, the features are mapped to the specified dimensionality by a linear map.

After processing the resulting node features through the graph transformer (Shi et al., 2021) with 'num_layers_enc'-many layers, they are aggregated in different ways and mapped to specified dimensions as defined by 'enc_<>_dim' for the atoms, scaffolds, join nodes, and leaf nodes, respectively. On top, the scaffold embeddings are additionally processed with the same transformer architecture ('num_layers_scaffold_enc') to inform the embedding about the scaffold-level connectivity. We then concatenate the resulting graph-level embeddings and further combine them with global molecule features, again computed via RDKit and then mapped to the required dimension ('enc_global_dim'), before mapping them to the latent space via the latent module which has 'num_layers_latent'-many layers.

Table 3: Parameter configuration of the best MAGNet runs.

| | Parameter | Value | | Parameter | Value |
|---|---|---|---|---|---|
| **Train** | batch_size | 64 | **Model** | node_aggregation | sum |
| | flow_batch_size | 1024 | | num_layers_latent | 2 |
| | lr | $3.07 \times 10^{-4}$ | | num_layers_enc | 2 |
| | lr_sch_decay | 0.9801 | | num_layers_scaffold_enc | 4 |
| | flow_lr | $1 \times 10^{-3}$ | | num_layers_hgraph | 3 |
| | flow_lr_sch_decay | 0.99 | | | |
| | flow_patience | 13 | loss_weights | joins | 1 |
| | gradclip | 3 | | leaves | 1 |
| **Model** | latent_dim | 100 | | motifs | 1 |
| | enc_atom_dim | 25 | | hypergraph | 1 |
| | enc_scaffolds_dim | 25 | | | |
| | enc_joins_dim | 25 | beta_annealing | max | 0.01 |
| | enc_leaves_dim | 25 | | init | 0 |
| | enc_global_dim | 25 | | step | 0.0005 |
| | atom_id_dim | 25 | | every | 2500 |
| dim_config | atom_charge_dim | 10 | | start | 2000 |
| | atom_multiplicity_dim | 10 | | | |
| | scaffold_id_dim | 35 | | | |
| | scaffold_multiplicity_dim | 10 | | | |
| | motif_feat_dim | 50 | | | |
| | scaffold_hidden | 256 | | | |
| | scaffold_gnn_dim | 128 | | | |
| | motif_seq_pos_dim | 15 | | | |
| | leaf_hidden | 256 | | | |
| | latent_flow_hidden | 512 | | | |

### B.2 DECODER

All decoding steps are conditioned on the latent code $z$. From $z$, MAGNet employs two transformer decoder layers to autoregressively decode the set of scaffolds $\mathcal{S}$ by selecting tokens $S_i$ from the extracted scaffold vocabulary. Generation of the variable-sized multiset $\mathcal{S}$ ends with selecting a stop token. In the next step, an MLP predicts the connectivity $A$ matrix between individual nodes in a permutation-invariant manner. This prediction is solely based on the learnable scaffold token and multiplicity embeddings. Indicating multiplicity is required, as multiple scaffolds in $\mathcal{S}$ share the same scaffold type but have to be connected in different ways.

At this point, MAGNet instantiates the atom-level graph by expanding scaffold tokens to their untyped graph objects. These graphs without features are then first assigned atom types by transformer decoder layers. Subsequently, the atom features and the respective scaffold embeddings are used by an MLP

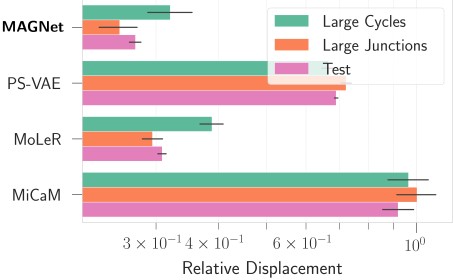

Figure 7: Displacement between latent representation of the input vs. the decoded output.

to assign bond types independently for every edge, thus creating $\mathcal{M}$. For any connection $A_{ij}$ between two motifs $M_i$ and $M_j$, another MLP then determines the join matrix $J^{(k,l)}$, that is used to identify the shared join node which is then "collapsed". This process applies only to atoms of the same type, is subject to valency constraints, and has to adhere to the predicted join node type $A_{ij}$.

After constructing the core molecule $\mathcal{C}$, MAGNet creates meaningful node embeddings by employing a graph neural network on the core molecule. These embeddings are the basis for a final module consisting of transformer decoder layers that equips the core molecule's atoms with leaf nodes $\mathcal{L}$. By the definition of the leaf nodes, every core molecule's atom can only have one leaf node. A leaf node prediction includes the node's atom type and the bond type connecting it to its attachment atom in the core molecule. The predicted bond type is again subject to valency constraints.

### B.3   ANALYSIS OF ACTIVE UNITS IN THE LATENT SPACE

Formally, the VAE optimises the ELBO

$$L = \mathbb{E}_{z \sim \mathbb{Q}}\big[\mathbb{P}\big(\mathcal{G} \mid z\big)\big] + \beta D_{\text{KL}}\big(\mathbb{Q}(z \mid \mathcal{G}) \mid P\big) \qquad \text{with} \quad P \sim \mathcal{N}(0, \mathbb{1})$$

where the posterior $\mathbb{Q}(z \mid \mathcal{G})$ is regularised towards the Normal prior $P$. In practice, finding a balance between the reconstruction loss $\mathbb{E}_{z \sim \mathbb{Q}}\big[\mathbb{P}\big(\mathcal{G} \mid z\big)\big]$ and the KL-divergence $D_{\text{KL}}$ is challenging. Yeung et al. (2017) and Burda et al. (2015) observe that optimising this objective can result in the VAE learning to collapse several units to the prior to compensate for few non-Gaussian components that support reconstruction. Behaviour like this can be measured through the number of active units in the latent space, defined as $\text{Cov}_{\mathcal{G}}\big(\mathbb{E}_{z \sim \mathbb{Q}(z \mid \mathcal{G})}[z]\big) > 0.02$ (Burda et al., 2015).

Due to generating the entire molecular context at each generation step, MAGNet heavily relies on the latent representation; also, our reconstruction experiments Sec. 5.1 support this. However, this intended behaviour requires the approximate posterior $\mathbb{Q}(z \mid \mathcal{G})$ to be close to the Normal prior $P$ to allow for good-quality samples. Although there are several methods available to improve the alignment between the approximate posterior and the prior, such as latent dropout (Yeung et al., 2017), a cyclic $\beta$-annealing schedule (Fu et al., 2019), and the GECO loss (Rezende & Viola, 2018), none of them have been able to achieve a rate of active units over 50 % beyond a simple weighting of the $D_{\text{KL}}$ term. As a result of this analysis, we fitted a normalising flow to the VAE, which was trained with low KL regularisation. For this, we follow the framework of Conditional Flow Matching (Tong et al., 2024; Lipman et al., 2023) and achieve 100 % active units.

### B.4   MAGNET: HYPERPARAMETERS AND TRAINING

Training MAGNet for one epoch takes around 30 minutes on a single 'NVIDIA GeForce GTX 1080 Ti'. We trained MAGNet for 30 epochs and fitted the latent normalising flow post-hoc for 5000 epochs in total and conducted a random hyperparameter sweep including the learning rate, beta annealing scheme, and the number of layers for the encoder and latent module. The MAGNet model reported in the main text has 12.6 M parameters and its configuration is depicted in Tab. 3. In its

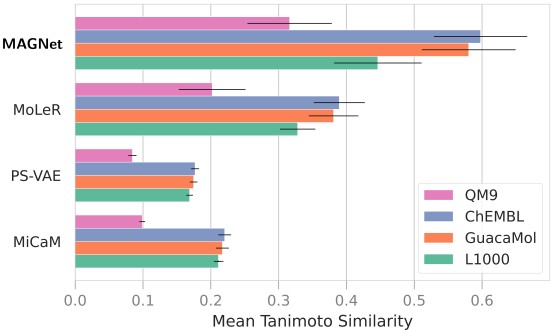

Figure 8: Additional quantitative evaluation shows that MAGNet is faithful to its latent code. From this follows decoding consistency even in challenging cases and on unseen datasets.

current version, MAGNet processes roughly 70 molecules per second during training and samples about 8 molecules per second during inference.

## C  DISCUSSION ON GENERATION APPROACHES

Zhu et al. (2022) categorise graph generation approaches into One-Shot and Sequential models based on how the latent code is mapped to the final graph. Models like GraphAF clearly fall into the One-Shot category, while MoLeR fits within the Sequential category. However, models such as DiGress, PS-VAE, and MAGNet do not match this classification perfectly.

According to Zhu et al. (2022), sequential models "generate a graph consecutively in a few steps," conditioning on a partial graph, i.e. complete parts of a molecule, at each step. Importantly, PS-VAE and MAGNet do not follow this scheme. Even though they generate a set of motifs autoregressively using an RNN or Transformer, they never generate a graph conditioned on a partial structure.

One-Shot models, as defined by Zhu et al. (2022), "generate a new graph represented in an adjacency matrix with optional node and edge features in one single step," first generating the set of node features and subsequently the set of edge features. This approach aligns more closely with the strategies employed by PS-VAE and MAGNet. The discrepancy arises due to the autoregressive manner in which these models generate the set of motifs.

Parallel to Zhu et al. (2022), Yang et al. (2022) propose similar terminology, categorising models into All-At-Once, Fragment-based (attaching fragments to a partial molecule), and node-by-node (attaching nodes to a partial molecule) methods. Under this classification, DiGress is considered an All-At-Once model, while PS-VAE is categorised as a Fragment-based method. This is inconsistent given that PS-VAE does connect any fragments until the final prediction step. Therefore, both MAGNet and PS-VAE would more appropriately fall into the All-At-Once category under this definition, while relying on a fragment-based or scaffold-based vocabulary.

To address these inconsistencies, we classify PS-VAE, MAGNet, and DiGress as One-Shot models, since they do not condition on a partial graph during generation (with diffusion models conditioning on a full but noisy graph). In each generation step, the module that generates motifs is invoked only once and does not depend on other modules like the bond-generating component.

## D  ADDITIONAL EXPERIMENTS

### D.1  ABLATION ON LARGER VOCABULARY SIZE

In the experiments conducted in Fig. 3c and b, we compare MAGNet to the top-$k$ vocabulary models MoLeR and PS-VAE. To ensure a fair comparison, we limit all models to a vocabulary size of 350. We conclude from this experiment, as well as e.g. Fig. 3a, that models cannot generate uncommon scaffolds, if they are not in their motif vocabulary and need to resort to constructing them from single

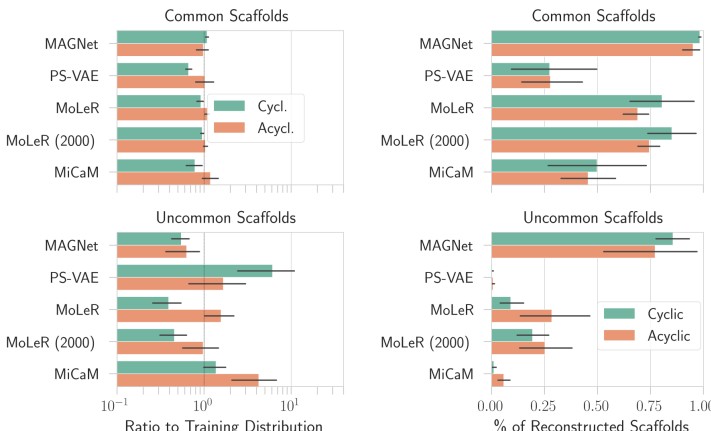

Figure 9: Left: percentage of reconstructed scaffolds. Right: comparison of sampled scaffolds to their frequency in the training distribution. A ratio of 1 is optimal.

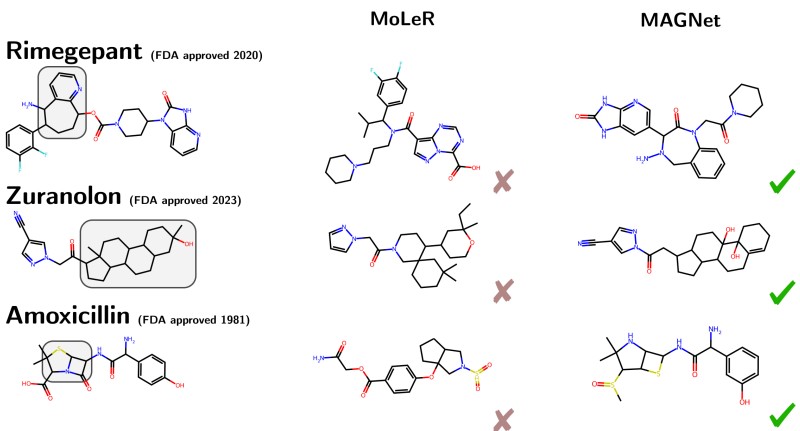

Figure 10: Qualitative examples of the reconstruction of FDA-approved drug-like molecules.

atoms. To investigate whether a model with just a significantly larger vocabulary can alleviate these shortcomings, we additionally conduct analysis with the MoLeR vocabulary trained with a vocabulary of size 2000 in Fig. 9. Interestingly, we can observe that a larger vocabulary does help the MoLeR model in sampling more uncommon scaffolds in better accordance with the training distribution. However, when investigating the model's ability to reconstruct uncommon scaffolds, even MoLeR with a larger vocabulary falls short. This indicates that uncommon scaffolds might be sampled at random during unconditional generation but the vocabulary becomes too large to navigate in the case of an explicit signal to decode a certain scaffold.

## D.2 DECODING DRUG-LIKE MOLECULES

Following up on the limitations of current molecular generators discussed in Fig. 1, Fig. 10 demonstrates that MAGNet does not display the same difficulties in decoding complex scaffolds.

## D.3 FULL BENCHMARK RESULTS

In addition to the results of the GuacaMol distribution learning benchmark results in Tab. 1, we provide the full benchmark results in Tab. 4.

Table 4: Metrics obtained by the MOSES and GuacaMol benchmarks for all evaluated models. We highlight the best method in each category.

| | CharVAE | SM-LSTM | GraphAF | HierVAE | MiCaM | JTVAE | MoLeR | PSVAE | MAGNet | DiGress |
|---|---|---|---|---|---|---|---|---|---|---|
| FCD/T ($\downarrow$) | 18.9 ± 16.0 | 0.50 ± 0.01 | 14.9 ± 0.34 | 3.52 ± 1.25 | 2.49 ± 0.16 | 1.67 ± 0.02 | 1.28 ± 0.05 | 6.57 ± 0.22 | 1.50 ± 0.00 | 2.79 ± 0.16 |
| SNN/T ($\uparrow$) | 0.29 ± 0.04 | 0.40 ± 0.00 | 0.25 ± 0.00 | 0.36 ± 0.03 | 0.34 ± 0.00 | 0.34 ± 0.00 | 0.36 ± 0.00 | 0.28 ± 0.00 | 0.34 ± 0.00 | 0.32 ± 0.00 |
| Frag/T ($\uparrow$) | 0.85 ± 0.13 | 1.00 ± 0.00 | 0.84 ± 0.03 | 0.96 ± 0.02 | 0.97 ± 0.01 | 0.98 ± 0.00 | 0.98 ± 0.00 | 0.78 ± 0.01 | 0.99 ± 0.00 | 0.97 ± 0.00 |
| Scaf/T ($\uparrow$) | 0.17 ± 0.15 | 0.38 ± 0.01 | 0.21 ± 0.03 | 0.37 ± 0.10 | 0.34 ± 0.01 | 0.24 ± 0.02 | 0.34 ± 0.02 | 0.20 ± 0.02 | 0.38 ± 0.01 | 0.25 ± 0.02 |
| IntDiv ($\uparrow$) | 0.85 ± 0.04 | 0.87 ± 0.00 | 0.93 ± 0.00 | 0.87 ± 0.01 | 0.87 ± 0.00 | 0.86 ± 0.00 | 0.87 ± 0.00 | 0.89 ± 0.00 | 0.88 ± 0.00 | 0.87 ± 0.00 |
| IntDiv2 ($\uparrow$) | 0.83 ± 0.05 | 0.86 ± 0.00 | 0.91 ± 0.00 | 0.86 ± 0.01 | 0.87 ± 0.00 | 0.86 ± 0.00 | 0.86 ± 0.00 | 0.88 ± 0.00 | 0.87 ± 0.00 | 0.86 ± 0.00 |
| Filters ($\uparrow$) | 0.36 ± 0.13 | 0.59 ± 0.01 | 0.47 ± 0.03 | 0.58 ± 0.05 | 0.54 ± 0.02 | 0.56 ± 0.01 | 0.56 ± 0.00 | 0.86 ± 0.01 | 0.71 ± 0.00 | 0.66 ± 0.02 |
| logP ($\downarrow$) | 2.24 ± 2.41 | 0.12 ± 0.01 | 0.41 ± 0.02 | 0.36 ± 0.17 | 0.20 ± 0.05 | 0.28 ± 0.03 | 0.13 ± 0.02 | 0.34 ± 0.02 | 0.22 ± 0.01 | 0.40 ± 0.07 |
| SA ($\downarrow$) | 0.42 ± 0.08 | 0.04 ± 0.02 | 0.88 ± 0.10 | 0.20 ± 0.14 | 0.51 ± 0.03 | 0.34 ± 0.01 | 0.06 ± 0.01 | 1.18 ± 0.05 | 0.12 ± 0.01 | 0.38 ± 0.07 |
| QED ($\downarrow$) | 0.16 ± 0.16 | 0.00 ± 0.00 | 0.22 ± 0.01 | 0.03 ± 0.00 | 0.08 ± 0.00 | 0.01 ± 0.00 | 0.01 ± 0.01 | 0.05 ± 0.00 | 0.01 ± 0.00 | 0.03 ± 0.00 |
| weight ($\downarrow$) | 32.9 ± 6.74 | 2.45 ± 0.36 | 96.9 ± 4.99 | 18.6 ± 8.93 | 51.1 ± 6.58 | 2.90 ± 0.06 | 5.96 ± 1.16 | 38.3 ± 2.14 | 14.5 ± 0.43 | 27.5 ± 0.04 |
| Valid ($\uparrow$) | 0.09 ± 0.01 | 0.96 ± 0.01 | 1.00 ± 0.00 | 1.00 ± 0.00 | 1.00 ± 0.00 | 1.00 ± 0.00 | 1.00 ± 0.00 | 1.00 ± 0.00 | 1.00 ± 0.00 | 0.85 ± 0.01 |
| Unique ($\uparrow$) | 0.95 ± 0.07 | 1.00 ± 0.00 | 0.91 ± 0.01 | 0.96 ± 0.01 | 0.98 ± 0.00 | 1.00 ± 0.00 | 1.00 ± 0.00 | 1.00 ± 0.00 | 1.00 ± 0.00 | 0.99 ± 0.00 |
| Novelty ($\uparrow$) | 0.95 ± 0.07 | 0.98 ± 0.00 | 0.91 ± 0.01 | 0.96 ± 0.01 | 0.98 ± 0.00 | 1.00 ± 0.00 | 1.00 ± 0.00 | 1.00 ± 0.00 | 1.00 ± 0.00 | 0.99 ± 0.00 |
| KL div. ($\uparrow$) | 0.63 ± 0.26 | 1.00 ± 0.00 | 0.67 ± 0.01 | 0.92 ± 0.01 | 0.94 ± 0.00 | 0.94 ± 0.00 | 0.98 ± 0.00 | 0.83 ± 0.00 | 0.95 ± 0.00 | 0.91 ± 0.00 |
| FCD ($\uparrow$) | 0.11 ± 0.13 | 0.93 ± 0.00 | 0.05 ± 0.00 | 0.53 ± 0.14 | 0.63 ± 0.02 | 0.75 ± 0.00 | 0.80 ± 0.01 | 0.28 ± 0.01 | 0.76 ± 0.00 | 0.65 ± 0.00 |

To provide further details on the results presented in Tab. 2, we provide the achieved scores for all evaluated methods for each of the GuacaMol goal-directed benchmark targets in Tab. 5.

Table 5: Results for the individual tasks of the GuacaMol goal-directed benchmark.

| | Gradient Ascent | | | MSO | | |
|---|---|---|---|---|---|---|
| | MAGNET | MICAM | MOLER | MAGNET | MICAM | MOLER |
| Albuterol similarity | 0.508 | 0.425 | 0.485 | 0.725 | 0.563 | 0.775 |
| Aripiprazole similarity | 0.622 | 0.558 | 0.585 | 0.768 | 0.712 | 0.841 |
| Mestranol similarity | 0.362 | 0.372 | 0.307 | 0.443 | 0.458 | 0.469 |
| C11H24 | 0.389 | 0.369 | 0.348 | 0.816 | 0.634 | 0.671 |
| C7H8N2O2 | 0.360 | 0.388 | 0.372 | 0.837 | 0.536 | 0.935 |
| C9H10N2O2PFCl | 0.515 | 0.461 | 0.479 | 0.787 | 0.665 | 0.743 |
| CNS MPO | 0.992 | 0.984 | 0.994 | 1.000 | 1.000 | 1.000 |
| Cobimetinib MPO | 0.858 | 0.830 | 0.835 | 0.901 | 0.899 | 0.895 |
| Fexofenadine MPO | 0.734 | 0.759 | 0.758 | 0.814 | 0.786 | 0.821 |
| Osimertinib MPO | 0.832 | 0.810 | 0.831 | 0.855 | 0.852 | 0.868 |
| Physchem MPO | 0.652 | 0.598 | 0.573 | 0.813 | 0.725 | 0.796 |
| Pioglitazone MPO | 0.637 | 0.571 | 0.403 | 0.960 | 0.855 | 0.878 |
| Ranolazine MPO | 0.206 | 0.177 | 0.200 | 0.798 | 0.730 | 0.764 |
| Median molecules | 0.210 | 0.151 | 0.159 | 0.302 | 0.149 | 0.306 |
| Celecoxib rediscovery | 0.235 | 0.273 | 0.232 | 0.414 | 0.317 | 0.449 |
| Thiothixene rediscovery | 0.123 | 0.218 | 0.275 | 0.308 | 0.391 | 0.262 |
| Troglitazone rediscovery | 0.167 | 0.233 | 0.219 | 0.261 | 0.268 | 0.336 |
| QED | 0.907 | 0.899 | 0.916 | 0.946 | 0.946 | 0.948 |
| TPSA target: 150.0 | 0.821 | 0.868 | 0.894 | 1.000 | 0.984 | 1.000 |
| logP target: $-1.0$ | 0.954 | 0.887 | 0.935 | 1.000 | 0.995 | 1.000 |
| logP target: 8.0 | 0.636 | 0.723 | 0.591 | 1.000 | 0.923 | 0.993 |

## D.4 DISPLACEMENT OF LATENT CODES

To quantify the discrepancy between input and reconstructed molecule visible in Fig. 3a, we measure the displacement of latent codes in Fig. 7. That is, we obtain the latent representation for the input molecule, decode this latent representation into the output molecule and then obtain the latent representation for the output molecule. This verifies what can be observed qualitatively in Fig. 3a–the evaluated baselines can not reliably decode complex scaffolds.

## D.5 INTERPOLATION

Extending on Fig. 5, we additionally provide examples for latent space interpolation in Fig. 12. During interpolation, MAGNet stays faithful to the scaffolds present in the input molecules. The last row shows a failure case of MAGNet: it identifies a scaffold multiset that can not be fully connected to a molecule.

Table 6: GuacaMol and MOSES Benchmark for ablations of MAGNet

| | FCD ($\uparrow$) | KL ($\uparrow$) | IntDiv ($\uparrow$) | logP ($\downarrow$) | SA ($\downarrow$) | QED ($\downarrow$) |
|---|---|---|---|---|---|---|
| MAGNet | 0.76 ± 0.00 | 0.95 ± 0.00 | 0.88 ± 0.00 | 0.22 ± 0.01 | 0.12 ± 0.01 | 0.01 ± 0.00 |
| no NF | 0.65 ± 0.00 | 0.92 ± 0.00 | 0.88 ± 0.00 | 0.38 ± 0.06 | 0.24 ± 0.03 | 0.01 ± 0.00 |
| Binary $A$ | 0.66 ± 0.00 | 0.92 ± 0.00 | 0.89 ± 0.00 | 0.43 ± 0.05 | 0.28 ± 0.02 | 0.04 ± 0.00 |

## D.6 TRANSFERABILITY OF SCAFFOLDS VIA ZERO-SHOT RECONSTRUCTION

We calculate the Tanimoto similarity in the reconstruction setting for a variety of datasets, Fig. 8. For all evaluated datasets, MAGNet achieves the best similarity scores between molecules, highlighting the transferability of scaffolds across various distributions.

We compute the Tanimoto scores only for those molecules that can be represented via the scaffolds that were extracted from the ZINC dataset. For the QM9 dataset, MAGNet can represent roughly 75% of the molecules in the dataset. This is due to unseen scaffolds which make up around 11% out of the total number of 289,966 scaffolds. For GuacaMol, MAGNet can represent around 97% of the molecules in the dataset. Out of the 9,562,028 scaffolds in GuacaMol, only 0.5% are missing from the scaffolds vocabulary extracted from the ZINC dataset. We consider a fragmentation into scaffolds that is more flexible and translates even better across datasets important future work.

## D.7 MAGNET ABLATION STUDIES

We show additional results for ablations of different parts of the MAGNet model in Tab. 6, performing the same analysis as done in Sec. 5.2. MAGNet without a normalising flow achieves an FCD score of 0.65, leading to a performance decrease of more than 14%. A similar decrease can be observed for MAGNet with only a binary scaffold connectivity $A$. This result further verifies that the atom type of a join node $j$ is an important conditioning for the generation of motifs $\mathcal{M}$.

## D.8 HALLUCINATING SCAFFOLDS

In Fig. 3 we compare the ratio between generated scaffolds and scaffolds in the dataset. However, often, the evaluated models generate scaffolds that never appear in the dataset. We refer to this behaviour as "hallucination" and quantify this observation in Fig. 11. We can observe that both PS-VAE and MiCaM often generate scaffolds that are not part of the training data and sometimes also MoLeR constructs out-of-distribution scaffolds via single atoms. By design, MAGNet samples in distribution

## D.9 GOAL-DIRECTED MOLECULE GENERATION

We evaluate MAGNet as well as our baseline generative methods on the GuacaMol goal-directed benchmark Brown et al. (2019). It aims to test a model's ability to explore the chemical space

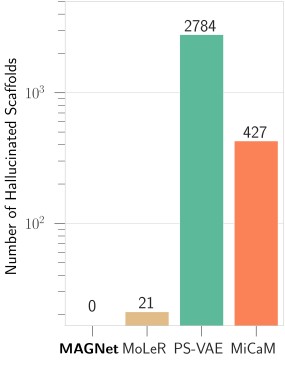

Figure 11: Number of scaffolds that were generated at sampling time but do not appear in the dataset.

Table 7: Additional results for the GuacaMol goal-directed benchmarks. All scores should be maximised. We report mean and standard deviation and highlight the best method in **bold**.

| | MAGNet | MoLeR | MiCaM | Random |
|---|---|---|---|---|
| Isomers | $\mathbf{0.81 \pm 0.00}$ | $\mathbf{0.81 \pm 0.02}$ | $0.56 \pm 0.01$ | $0.17 \pm 0.12$ |
| MPO | $\mathbf{0.89 \pm 0.01}$ | $0.88 \pm 0.01$ | $0.85 \pm 0.01$ | $0.61 \pm 0.22$ |
| Property | $\mathbf{0.99 \pm 0.00}$ | $\mathbf{0.99 \pm 0.00}$ | $0.93 \pm 0.00$ | $0.34 \pm 0.30$ |
| Redisc. | $0.33 \pm 0.01$ | $\mathbf{0.35 \pm 0.01}$ | $0.33 \pm 0.00$ | $0.14 \pm 0.01$ |
| Similarity | $0.56 \pm 0.05$ | $\mathbf{0.60 \pm 0.06}$ | $0.47 \pm 0.06$ | $0.26 \pm 0.08$ |

through a variety of single and multi-objective optimization tasks. We employ two latent optimisation mechanisms and evaluate all models under the same setting. In Tab. 2, we train a proxy regressor consisting of a simple MLP with default parameters Pedregosa et al. (2011) for each of the required score functions on a subset of $10,000$ labelled samples. The proxy regressor maps from latent space embedding to the predicted score for that embedding. During optimisation, we then utilise a gradient ascent procedure to find a latent code with a high predicted score. Across models, we use $100$ gradient steps per sample with a learning rate of $0.01$. We consider this latent procedure an especially important means of evaluating the generative models, as it indicates to what extent the latent space is organised smoothly with respect to certain properties.

Additionally, we also employ "Molecular Swarm Optimisation" as a second technique for goal-directed generation Winter et al. (2019). We use a configuration of one swarm per sample initialised with $50$ particles and we run the optimisation procedure for $10$ rounds. We report these results in Tab. 7. We can observe from these results that especially MoLeR benefits from the random noise introduced by this latent procedure. Nevertheless, MAGNet performs competitively or better on almost all evaluated tasks.

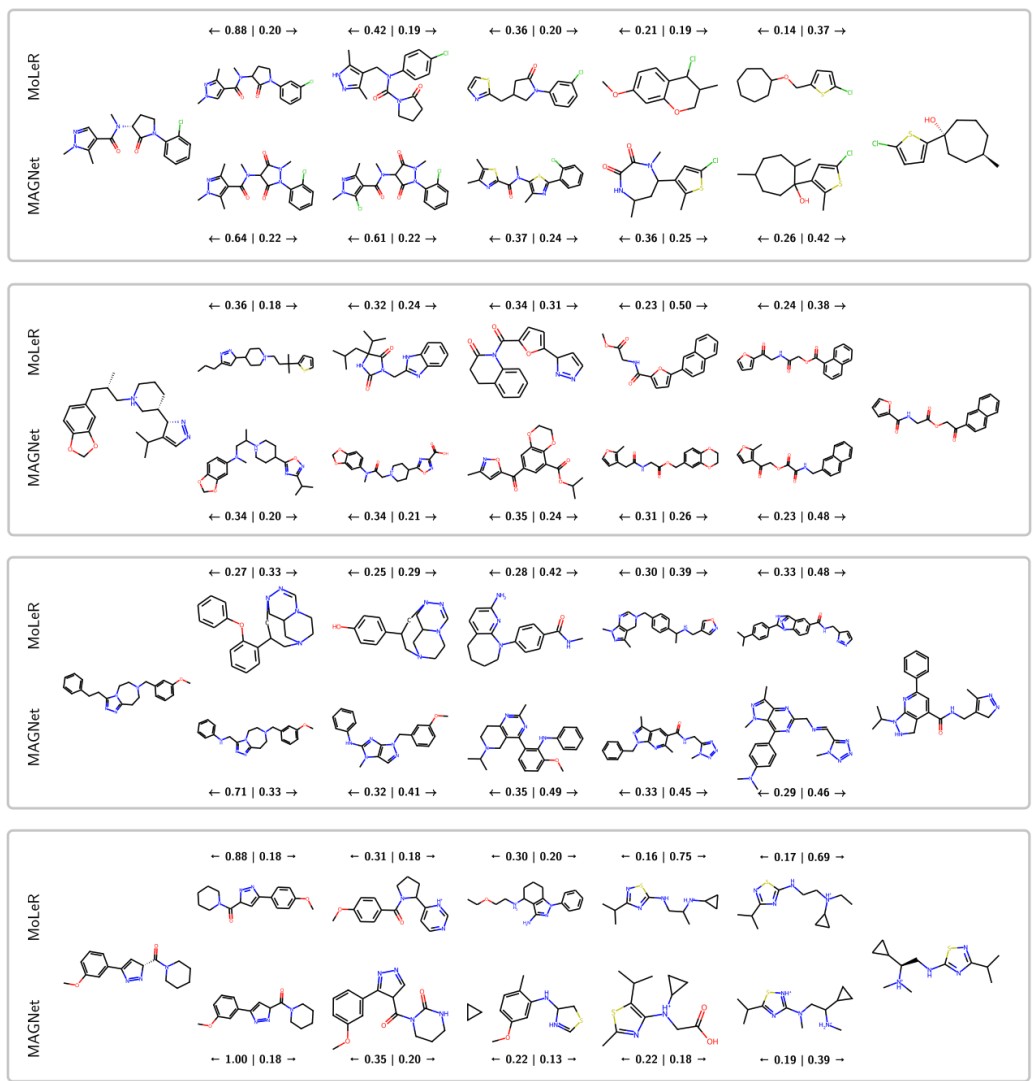

Figure 12: We provide four interpolation examples for MAGNet and MoLeR. The input molecules (left and right) are shared between the two models. We report the Tanimoto similarity as a rough estimate for the interpolation's goodness.

