# OpenReview forum: "MAGNet: Motif-Agnostic Generation of Molecules from Scaffolds"
_ICLR.cc/2025/Conference — ICLR 2025 Spotlight_

### Official Review · Reviewer_8uqV · 2024-10-30

**Soundness:** 2
**Presentation:** 2
**Contribution:** 2
**Rating:** 5
**Confidence:** 4

**Summary:**

This paper addresses a critical limitation in substructure-based molecular generative models: the inability to capture the structural diversity in molecular space due to missing complex structures from the motif vocabulary. The authors propose a novel approach that employs a structural scaffold vocabulary, leaving atom and bond types to be predicted by the model. This approach is intended to enrich structural diversity, with specific metrics introduced to highlight the advantages of the proposed method.

**Strengths:**

1. The paper tackles a significant issue in 2D molecular generation, and Figure 1a provides an illustrative example that emphasizes the limitations of current motif-based methods in generating novel molecular structures. This emphasis on structural diversity is both timely and impactful for the field.


2. Experimental results demonstrate an increase in generated structural diversity.

**Weaknesses:**

1. **Limitations in Generating Novel Molecular Structures**

1.1 A primary concern is whether the proposed solution truly resolves the problem of generating novel molecular structures. As highlighted in Figure 1a, the generation of new substructures—particularly complex fused ring systems—appears challenging. Both the original motif approach and the scaffold motif proposed in this work seem constrained in their ability to construct unseen complex fused rings because they are hard to piece together using substructures in the vocabulary.

1.2 Furthermore, while the scaffold motif approach can potentially reduce vocabulary size, it still requires additional specification of bonds and atom types on rings, which might affect the validity of atom valence of the generated molecules.

Thus, while the problem raised is compelling, the method appears to partially address, rather than fully resolve, this issue.

2. **Writing**

2.1 Overlap in Sections 3 and 4: Sections 3 and 4 present overlapping content, which could benefit from a clearer delineation. Specifically, Section 4 should focus more on detailing the network structures (architecture) within each module, and provide an illustration of the model structure. The high-level probability descriptions, already covered in Section 3, could be streamlined here.

2.2 Figure Captions: In Figure 3, the captions for panels (b) and (c) appear reversed.

2.3 References: The authors should thoroughly review the reference list for consistency. It is full of preprint references.  Accepted papers should reference their final publication locations rather than preprints (e.g. MiCaM and ShapeMol, etc). Please also check with repeated links and unusual endings with paper abbreviations (e.g. G-SchNet, JODO, etc.).

3. **Experimental Results**

While structural diversity has been enhanced, some important metrics, such as FCD and SA, appear to drop. Additionally, there is no mention of the validity rate of generated molecules in the table 1.

**Questions:**

1.
Could the authors provide the **validity** of generated molecules? Are there specific metrics used to ensure that generated molecules are fully **connected**, given that some examples (such as in the bottom row of Figure 11) suggest molecules with broken structures?

2.
A comparison with baseline methods in terms of **generation speed** would add valuable context for practical deployment.

3.
Could the authors clarify the **vocabulary size** difference between your approach and baseline models?

---

> ### Author Response · Authors · 2024-11-20
>
> &nbsp;
>
> We want to thank the reviewer for their thoughtful review. We particularly appreciate your recognition of the significance of the problem we address and our contribution in highlighting the limitations of current motif-based approaches. We provide detailed answers to the raised questions and concerns in the following.
>
> &nbsp;
>
> > Is the problem of structural diversity fully resolved? & Significance of experimental results
> - We believe that making the community aware of critical limitations of fragment-based methods together with our novel method and evaluation are important contributions and we hope that our work will spark followup research that will eventually resolve the challenges regarding structural diversity.
> - Problem and Evaluation: In our work, we propose a new perspective on the evaluation of molecular generative methods. While standard benchmarks rely heavily on metrics like FCD scores, our work demonstrates that this focus is insufficient. A similar observation has also been made in the context of computer vision [1]. Accordingly, we introduce novel evaluation criteria to account for the importance of structural expressivity, which is an overlooked aspect of molecular generation.
> - Approach: Our scaffold-motif approach represents a step towards addressing these limitations, even if it does not fully resolve them. The drop in standard metrics like FCD and SA actually highlights an important trade-off: methods optimised for these conventional metrics sacrifice structural expressivity. With MAGNet, we aim to strike a balance that is overall beneficial for molecule generation and, thus, drug discovery.
>
> > Impact on separation of structure and features on validity of generated samples, Q: Validity and Connectivity
> - We agree that the impact of the changed inductive bias has to be analysed thoroughly.
>     - For the effect on sample quality, we specifically dedicated our analysis in Sec.5.3 to the accurate prediction of features both in terms of generation (Fig.4 a,b) and reconstruction (Fig.4c). We clearly demonstrate that the free featurisation as implemented through MAGNet is not a limitation but a feature and beneficial for sample diversity.
>     - We want to further support these findings by referring the reviewer to App. D.3, Tab.4 in particular, which complements the benchmark from Sec. 5.2. Unfortunately, we missed to make this reference in the initial submission and have updated the manuscript accordingly, cf. L419. Due to validity constraints, MAGNet achieves 100% validity. Like other methods, we always consider the largest connected component for analyses. For the complete generated output, as shown in Fig.12, the retention percentage for MAGNet is >93%, with 80% of the generated molecules being fully connected.
>
> > Writing
> - We structured the manuscript to separate the general mathematical framework of our factorisation (Sec. 3) from the model (MAGNet) that matches this factorisation (Sec. 4). Following the reviewer’s feedback, we have clarified the manuscript structure with an introductory note in Section 3 and detailed the network architectures throughout Section 4, making the distinction between theoretical framework and implementation more explicit.
> - Many thanks for pointing out the incorrect panel order in Fig.3. We have corrected this in the updated pdf.
> - We have updated all references to cite the published versions of the papers and revised the manuscript accordingly.
>
> > Q: Generation speed:
> - We conducted a quantitative analysis on the generation speed. While the reported results certainly reflect aspects of the methodological design, the specific generation times also heavily depend on the chosen framework and implementation. However, our choice to design MAGNet in a hierarchical OS fashion relies more on the advantages connected to its ability to consider the entire molecular context at all generation steps. Please refer to our response to reviewer hxc4 (W1) for additional details.
> |Method|s/10 samples|
> |-|-|
> |PS-VAE|0.293|
> |MiCaM|0.583|
> |MoLeR|2.760|
> |MAGNet|1.200|
>
> > Q: Vocabulary Size:
> - Using our fragmentation approach, the vocabulary reduces to 347 distinct scaffolds, which corresponds to 7371 motifs. Note that our fragmentation is more effcient than those used in MoLeR and MiCaM, which amount to roughly 14000 and 15600 motifs, respectively.
> - For a fair comparison, we chose the same vocabulary size (350) for all baselines. Inspired by the question of reviewer w2Yn, we also compare MAGNet against a MoLeR model with a vocabulary size of 2k, which we discuss in our general response.
>
> &nbsp;
>
> We hope that we have adequately addressed the raised concerns and believe the reviewer's feedback has enhanced the quality of our manuscript. We look forward to further discussion.
>
> &nbsp;
>
> [1] Jiralerspong et al., Feature Likelihood Score: Evaluating Generalization of Generative Models Using Samples, 2023

---

> > ### Comment · Reviewer_8uqV · 2024-11-22
> >
> > Thank you for your response. However, I still feel that my main concern has not been addressed. Based on my understanding, your method still struggles to generate complex fused ring systems as shown in Figure 1a, correct? These systems involve not only innovative atom types on common rings but also novel innovations on the scaffold itself. As you mentioned in your motivation, some known drugs feature such unique scaffolds. This is something that traditional motif-based methods cannot easily generate from their vocabulary, and it seems that your scaffold vocabulary also fails to recover these types of scaffolds. So, while you have raised an important issue, it appears that the method has not yet effectively solved the problem you highlighted.

---

> ### Author Response · Authors · 2024-11-22
>
> &nbsp;
>
> Thank you for the prompt response and engaging in a discussion with us. We believe there has been a fundamental misconception regarding Fig. 1 and apologise that we misunderstood the reviewer's main concern. In the following, we would like to clarify this essential aspect of our work.
>
> &nbsp;
>
> > Based on my understanding, your method still struggles to generate complex fused ring systems as shown in Figure 1a, correct?
>
> - That is indeed not correct. **Quite the opposite, with our approach complex scaffolds such as fused rings become accessible**.
> - In Fig.1a we only highlight the shortcomings of existing motif-based methods (in particular MoLeR) on the reconstruction of approved drugs with complex scaffolds. This presentation was meant to motivate the necessity for rethinking motif-based molecular generators.
> - In App.D.2 (Figure 10), we demonstrate how MAGNet *can* decode the complex scaffolds shown in Fig.1a.
> - We also explicitly evaluate the decoding capabilities of such scaffolds quantitatively in Sec.5.1, Fig.3 in particular. **What is referred to as uncommon cyclic scaffolds are exactly those complex scaffolds the reviewer mentions and we improve upon baselines significantly**.
> - It is important to highlight that our vocabulary is much more expressive than existing ones, containing scaffolds of structures like Pentacene, Triphenylene and Benzo[a]pirene, and generally scaffolds of fused rings with up to 6 rings, as well as bicyclic scaffolds.
> - **To avoid future misunderstandings and if the reviewers finds this beneficial, we could imagine to swap the presentations of Fig.1a and Fig.10**. This change would aid in showing what MAGNet can do in the main part of the manuscript and highlight limitations of existing works only in the appendix.
>
> &nbsp;
>
> We sincerely hope that this clears up the misunderstanding regarding MAGNet’s capabilities and our contributions. In light of this clarification, we would appreciate if the reviewer reconsiders their evaluation and look forward to further discussion.

---

> > ### Comment · Reviewer_8uqV · 2024-11-24
> >
> > Thank you for your detailed response and for providing additional results. I agree that swapping the presentations of Fig. 1a and Fig. 10 could potentially highlight your contributions more effectively.
> >
> > But, I still find it unclear why your vocabulary is considered more expressive and better equipped to accommodate more fused rings. From my perspective, abstracting motifs to scaffolds does not seem to address this challenge directly. Could you kindly provide a more intuitive explanation?

---

> > > ### Author Response · Authors · 2024-11-24
> > >
> > > &nbsp;
> > >
> > > Thank you for your follow-up and further clarifying your concern. We appreciate the opportunity to further elaborate on how MAGNet addresses the challenge of generating complex fused ring systems.
> > >
> > > &nbsp;
> > >
> > > In our work, we show that generative models operate with a fixed vocabulary, effectively limiting the number of “spots” available to represent structural components such as fused rings:
> > > - **Models fail at generating complex structures, such as fused systems, that are not present in the vocabulary**. We demonstrate that the reliance on single atom representations in such scenarios is insufficient, cf. Figure 3 and 4.
> > > - Moreover, **the naive approach of expanding the vocabulary size is not adequate.** We show that approaches like MoLeR (2000) and MiCaM cannot leverage their significantly larger vocabularies to address the challenge of generating complex fused ring systems, cf. Figure 9.
> > >
> > > &nbsp;
> > >
> > > We conclude that **it is essential to allocate the limited vocabulary capacity wisely**. We present a solution to this in our work:
> > > - Traditional motif-based models like MoLeR and PS-VAE allocate a disproportionate number of “spots” in their vocabularies to redundant scaffolds (from a structural perspective) with different motif representations, cf. Figure 6b. For example, slight variations in a single-ring structure can occupy multiple spots in their vocabulary, resulting in inefficiency.
> > > - MAGNet, on the other hand, takes a more targeted approach: it *learns* atom and bond types, freeing up vocabulary capacity to represent the structures that are truly challenging to learn—complex scaffolds such as fused rings and uncommon junctions.
> > > - **Because of its efficient use of vocabulary capacity, MAGNet can store and represent a broader range of fused ring systems and retains only the essential structural information in its vocabulary.**
> > > - Minor but still relevant, our process of building the vocabulary (fragmentation), cf. Section 4.1, is already more efficient than existing approaches (cf. previous answer) which further benefits MAGNet in generating complex structures.
> > >
> > > &nbsp;
> > >
> > > In summary, MAGNet’s design and the underlying expressive vocabulary enable the effective generation of complex fused ring systems. We sincerely hope this explanation clarifies why our method is inherently better equipped to address the mentioned challenge.

---

> > > > ### Comment · Reviewer_8uqV · 2024-11-29
> > > >
> > > > Thank you for your detailed response to my comments. I now have a clearer understanding of how your method constructs a more efficient vocabulary and accommodates complex fused ring structures. I appreciate the effort you have put into addressing my concerns.
> > > >
> > > > Based on this, I have raised my score from 3 to 5. However, I could not provide a higher score as I feel that the experimental results are not yet fully competitive; for instance, metrics such as FCD and SA show some decline compared to certain baselines.
> > > >
> > > > Best regards,
> > > >
> > > > Reviewer

---

> > > > > ### Author Response · Authors · 2024-12-01
> > > > >
> > > > > &nbsp;
> > > > >
> > > > > We thank the reviewer for their engagement and are **pleased that we successfully addressed their main concern about how MAGNet generates complex structures, such as fused rings**. We appreciate that this is reflected in an increased score.
> > > > >
> > > > > &nbsp;
> > > > >
> > > > > Furthermore, we acknowledge the reviewer’s criticism regarding MAGNet’s performance in some metrics, such as FCD or SA. We would like to take the chance and put it in the global context of our work:
> > > > >
> > > > > - In our manuscript, we emphasise the insufficiency of metrics like FCD, a perspective supported by literature in Computer Vision. Our work provides a new approach to enhance structural expressivity, which cannot be adequately captured by FCD scores, as demonstrated in our analysis in Sec.5.2, cf. L425 ff.
> > > > > - Singling out the FCD metric conveys an incomplete picture of our work. We demonstrate how MAGNet improves upon baselines throughout our experiments, cf. Figs.1/3/4.
> > > > > - Finally, we do not consider the drop in the SA score a significant limitation but our proposed solution regarding synthesizabilty, as discussed in the limitation section (L285 ff.), more relevant for practical scenarios.
> > > > >
> > > > > &nbsp;
> > > > >
> > > > > We are grateful for the constructive feedback and the discussion with the reviewer.
> > > > >
> > > > >
> > > > > Best regards,
> > > > > The Authors

---

### Official Review · Reviewer_w2Yn · 2024-10-31

**Soundness:** 4
**Presentation:** 3
**Contribution:** 3
**Rating:** 8
**Confidence:** 4

**Summary:**

In this work, the authors aim to address a fundamental limitation of fragment-based molecular generators. The vocabulary of such models is defined as the union of common molecular fragments, obtained from datasets of known molecules, and individual atoms, which can be used to re-generate motifs that would not be listed under the available fragments. The authors argue that this choice of vocabulary creates an inherent tradeoff in the expressivity of the model. On the on hand, including more fragments quickly increases the vocabulary of the model, with the number of motif variation increasing exponentially with their size. On the other hand, learning to model missing fragments from individual atoms is a challenging task requiring even more training data and often leading to unrealistic molecular motifs.

As a solution, the authors propose a coarse-to-fine-grained molecular generation paradigm centred around molecular scaffolds, rather than fragments, as the basic building block of the model's vocabulary. A single scaffold implicitly captures many similar fragments, allowing for a relatively small vocabulary size while retaining expressivity. The proposed model, MAGNet, a VAE-based molecular generator, operates on this multi-level factorisation paradigm, by first sampling scaffolds and only then specifying the atomic composition of the scaffolds, their joints and the leaf nodes as a successive step. They evaluate this approach on several benchmarks against a variety of baselines and show that the proposed method, with its more expressive vocabulary, can reliably generate complex molecular motifs, in addition to allow for latent code optimisation and interpolation.

**Strengths:**

In general, I found the paper interesting and very well written. It clearly identifies a limitation of current fragment-based molecular generators in terms of expressivity, and present a well motivated solution based on a novel factorisation paradigm. I believe this is a good paper which brings significant contributions to fragment-based molecular generation approaches and for this reason I recommend that the paper should be accepted. I suggested some clarifications (see below) which I think would increase the clarity of the paper and complement the discussion on the limitations of the method and the general positioning of fragment-based and one-shot graph molecular generators.

A few specific notes:
1. The Related Work section is exhaustive.
2. The proposed method is clearly introduced and detailed. In particular Figures 2 and 6 effectively convey the hierarchy supporting this paradigm.
3. The experimental section is well structured and the authors compare to a large array of prior work and using several public datasets. The results effectively support the main claims made in the paper. In particular, the results presented in Figure 3, Figure 4 are compelling.

**Weaknesses:**

1. One of the main limitation of fragment-based methods is the absence of synthesizability considerations in the framework design. This significantly limits the applicability of such methods since, to be tested in physical and biological assays, the proposed molecules either require individual and expansive custom synthesis plans or have to be replaced by available analogs, thus drastically under-utilizing the expressivity of the model. It would be interesting to further discuss the limitations of the proposed method w.r.t. synthesizability in the manuscript.
2. On line 371: "We do not report Novelty and Uniqueness, as almost all evaluated models achieve 100% on these metrics." if this claim is made, then the numbers should be presented (at least in Appendix). Same for the mention right after: "For the baselines DiGress, SM-LSTM, and CharVAE, which are not able to achieve 100% Validity, we sample until we obtain 10^4 valid molecules", it could be informative to include these numbers in appendix (validity rate for each method).
3. It would be useful to specify a bit more clearly how benchmarking on Guacamol is executed (lines 301-304 and 309-311). Is the model trained on ZINC and then evaluated on some reference set defined by Guacamol (Chembl)? Or is the model trained on Chembl molecules and compared to a test set also defined in Guacamol? While the provided references are useful, the description of the experiments should be standalone in the paper.
4. In the goal-directed evaluations, it would be interesting to compare MAGNet with methods specifically aimed at goal-directed molecular design such as RL and GFlownet based methods.

### Elements worth clarifying

1. Figure 1 could be made clearer by further explaining parts a), b) and c) in the caption.
2. The factorisation from graph to scaffold graph, and scaffold graph to molecular graphs, described in Section 3, would be clearer with a supporting Figure.
3. The main baselines (PS-VAE, MoLeR and MiCaM) could be described in greater details.
4. I found the discussion on novel conditioning capabilities very interesting (lines 469-476), however, even with this in mind, it is not clear to me what Figure 5 is showing or how it supports these claims. I think this figure could be improved to better support this discussion.

### Minor comments:

1. Typo on line 155, factorise*
2. Line 150: not sure that App C.6 is the intended link here (or how it relates to the sentence)?
3. Typo on line 302: to evaluate*
4. Error in Figure 3: based on the text and the caption itself, it seems to be that the graph columns B and C are mixed up in Figure 3.
5. I did not find the caption of Table 1 very natural to read. I would suggest simply specifying in parenthesis what underline and bold mean in the table, as opposed to underlining and bolding their description in the caption.

**Questions:**

1. My understanding of the experiments presented in Figure 1-a, Figure 3 and Figure 9 carried out (Section 5.1) is that the fragment-based approaches fail to reconstruct complex motifs that are absent from their fragment vocabulary by using atom-based tokens only. In contrast, MAGNet contained these structures in its scaffold vocabulary. Why haven't both methods used the same dataset to construct their fragment/scaffold vocabularies, without limiting fragment-based methods to only the top-K fragments (i.e. including all fragments). In this case, the modeling task from the fragment-based methods using a vast vocabulary of fragments would be more difficult, but the method wouldn't lack these important fragments such as big rings, preventing them from reconstructing specific molecules. Have experiments on the baselines been carried out by varying the value of "k" in the top-k fragment-based vocabularies to see how vocabulary size trades-off with learning complexity? It seems to me that this might be a point where a more appropriate tuning of the baseline's hyperparameters would make a difference. And if the main advantage of the proposed factorisation is that it removes the need for such tuning when constructing the fragment vocabulary, it would be interesting to discuss these considerations in the paper.

---

> ### Author Response · Authors · 2024-11-20
>
> &nbsp;
>
> We are grateful for the reviewer's insightful feedback and are pleased with the positive comments on our work. In the following, we would like to clarify the remaining questions and concerns.
>
> &nbsp;
>
> > Absence of synthesizability considerations
>
> - Thank you for this constructive feedback. We agree that synthesizability warrants discussion and have added a paragraph to the limitations section, cf. L285-288. While our manuscript focuses on standard small molecule datasets, we acknowledge that the absence of synthesizability considerations affects the practical application of fragment-based methods, particularly when moving from computational predictions to experimental validation. We believe that synthesizability is inherently linked to the training data, and that utilising a dataset of multi-component reactions [1] could help address this limitation.
>
> > Missing metrics (Novelty, Uniqueness, Validity)
>
> - We completely agree with the reviewer’s feedback and have missed to refer the reader to Appendix D.3, Tab. 4 in particular, which includes the mentioned metrics for Novelty and Uniqueness. Moreover, the number of samples to obtain 10k valid molecules can be deduced from the reported validity and amounts to roughly 12k for DiGress.
>
> > Clarification of the benchmarking procedure
>
> - As we outline in L321-322, all models are trained on the ZINC data and we conduct the GuacaMol and MOSES benchmarks on the respective test set of the ZINC data. We updated our manuscript to better reflect this setup, cf. L322. Note that we use the other mentioned datasets as part of the zero-shot experiment (Appendix D.6 and Figure 8) that investigates the ability of MAGNet and the baseline models to represent molecules from different distributions. Across datasets, we show that the inductive bias as set through our fragmentation and scaffold approach leads to more faithful reconstruction of molecules from their latent codes.
>
> > Inclusion of methods that specifically optimise for goal-directed generation of molecules
>
> - We appreciate this suggestion. In our manuscript, we focus on the comparison with fragment-based methods to establish a clear methodological context. For this, we delineate between the optimisation scheme (Gradient ascent and MSO) and the underlying generative procedure (MAGNet, MiCaM, MoLeR). Following the reviewer’s suggestion, we checked the setting used in [2]. Comparing our results for the goal-directed benchmark with those reported in [2], MAGNet should perform competitively. However, the goal-directed generation setting in our experiments and the results reported in [2] are not directly comparable due to the varying number of oracle calls. For the final version of the manuscript, we would be happy to include a thorough evaluation of MAGNet following the benchmark setting in [2].
>
> > Clarifications
> - We updated the caption of Fig.1 and aligned it better with the structure (a,b,c) of the figure as suggested by the reviewer.
> - In structuring the manuscript, we intended to distinguish between the general mathematical formulation of our factorisation approach (Section 3) and MAGNet's specific implementation details (Section 4), with Fig. 2 illustrating the key concepts of our factorisation, cf. L130-133. We would appreciate more specific guidance on which aspects are not sufficiently visualised in its current version.
> - Following the reviewer’s suggestions we tried to provide more specific information about the main baselines through the manuscript. In particular, we would like to refer the reviewer to L300-307.
> - Thank you for highlighting a lack of clarity for the conditioning setting presented in Fig. 5. In our response to Reviewer hxc4, we provide a more comprehensive discussion of the practical relevance and potential applications of this conditioning approach. We furthermore updated Sec.5.4 with these details to better illustrate how MAGNet enables flexible conditioning.
>
> > Minor Comments
> - We have addressed all minor comments and typos in the updated PDF. Thank you for thoroughly reviewing the manuscript and providing such detailed feedback.
>
> > Impact of vocabulary size and experiments analysing the effect
> - We thank the reviewer for their suggestion. We believe an analysis of this kind greatly supports the claims and contributions of our work and have included a detailed discussion in our general response to highlight these results for all reviewers.
>
> &nbsp;
>
> We appreciate the positive feedback and are excited to move forward with these improvements. We hope that we addressed all outstanding concerns satisfactorily and welcome further discussion.
>
> &nbsp;
>
> [1] Graziano et al.,  Multicomponent reaction-assisted drug discovery: A time-and cost-effective green approach speeding up identification and optimization of anticancer drugs, 2023
> [2] Gao et al., Sample Efficiency Matters: A Benchmark for Practical Molecular Optimization, 2022

---

> > ### Comment · Reviewer_w2Yn · 2024-11-21
> >
> > I want to acknowledge the response of the authors and I appreciate the efforts made to clarify certain elements of the paper and the addition of an additional experiments with larger vocabulary.

---

> > > ### Author Response · Authors · 2024-11-28
> > >
> > > We are grateful for your positive endorsement and detailed critique, which significantly enhanced the clarity and presentation of our research. We found the discussion with the reviewer a pleasant experience and are thankful for the improvements it brought for the final manuscript.

---

### Official Review · Reviewer_Ry7i · 2024-11-02

**Soundness:** 2
**Presentation:** 4
**Contribution:** 2
**Rating:** 8
**Confidence:** 4

**Summary:**

The paper presents MAGnet, a generative model for molecules. MAGnet is based on scaffolds (an abstraction of molecular fragments without atom and bond information, just the graph structure), which are introduced to factorize of the molecule distribution. MAGnet is a VAE-like architecture which generates new molecules by first predicting a scaffold set and its connectivity from latent space, then by predicting the atom and bond types for each scaffold, and finally by adding leaf nodes. Experiments show good performances in comparison with several baselines, on standard benchmarks such as GuacaMol and MOSES.

**Strengths:**

- using scaffolds instead of motifs qualifies as an innovative proposal, which could be of value to the community
- experimental evaluation is extensive in both depth (many baselines) and width (two benchmarks)

**Weaknesses:**

throughout the paper, MAGnet is considered a one-shot (OS) molecule generator, when in fact it is auto-regressive (AR). Appendix B.2 states that the set of scaffolds is generated auto-regressively, so I cannot understand how it could be considered one-shot. According to the definition by Zhu et al. (2022):


    Sequential generation refers to generating graphs in a set of consecutive steps, usually done nodes by nodes and edges by edges. One-shot generation, instead, refers to generating the node feature and edge feature matrices in one single step.

This is very different from what it is stated in Section 2 of the MAGnet paper:

    Zhu et al. (2022) categorise the generation process further into sequential methods, building molecules per fragment while conditioning on a partial molecule.

to justify the fact that MAGnet is OS. According to the same paper that is cited, MAGnet belongs to the AR category. If MAGnet is AR, then most claims made in the paper need to be toned down or changed because it falls in the same category as MoLeR, which has usually comparable or better performances than MAGnet. For example, the claim "MAGnet is the best OS generator" no longer makes sense.

**Questions:**

I'll be honest here. I really liked this paper and I was going to recommend clear acceptance until I understood that MAGnet was an AR model "disguised" as an OS model. From that point onward, I couldn't shake the feeling that it was framed as OS only because, if put in the AR category, the results would become not so impressive (although still good). I really hope this was an unintentional mistake or misinterpretation by the authors. That's a shame in my opinion, since I believe the proposal is original and the evaluation was very thorough, the technical side of this paper is almost flawless.

I will be recommending rejection of this paper for the moment, because in this form too many claims made in this paper stem from an incorrect premise. However, I am willing to hear from the authors why they consider MAGnet one-shot instead of sequential and will re-evaluate whether I could reconsider my judgment after the rebuttal phase.

---

> ### Author Response · Authors · 2024-11-20
>
> &nbsp;
>
> We thank the reviewer for their detailed feedback and thoughtful insights. We are grateful for the reviewer’s acknowledgement of scaffolds as an original and valuable proposal, as well as their recognition of the technical execution and soundness of our work. In the following, we would like to address the reviewer’s main concern: the categorisation of MAGNet as a one-shot (OS) model.
>
> &nbsp;
>
> > One-Shot (OS) terminology
> - In our manuscript, we aimed to adopt the established terminology to classify different methods. However, we acknowledge that the strict definitions do not perfectly apply to our approach, nor PS-VAE or DiGress. We sincerely appreciate the reviewer highlighting this.
> - Fundamentally, we argue that there is a difference between (i) the underlying neural network architecture used to generate aspects of the molecule and (ii) the process of molecule generation (sequential or one–shot). These two are orthogonal concepts in our view.
>     - Regarding (i), MAGNet and PS-VAE for example utilise, among others, recurrent and autoregressive architectures such as RNNs to generate specific, independent components of the molecule, e.g. the set of motifs.
>     - Regarding (ii), methods like MAGNet and PS-VAE divide the molecular graph into distinct parts, e.g. atom and bond types, which are predicted once and define different characteristics of the molecular graph. In contrast, sequential generation builds molecules step-by-step from parts that share the same characteristics, e.g. motifs or single atoms. This distinction is well reflected in the factorisations of these methods:
>         - MAGNET divides $P(G)$ into distinct components scaffolds $S$, their connectivity $A$ and representation $M$, joins $J$, and leaves $L$.
>         - Similarly, PS-VAE, factorises $P(G)$ into nodes (atoms and motifs) and bonds. We classify this method as OS for the same reason, even though it decodes e.g. the nodes with an autoregressive module.
>         - We also consider DiGress, while iterating over the graph generation in many steps, as an OS model, as it does not condition on a partial molecule over the generation, cf. L114-115.
>         - MoLeR, a sequential approach, generates the molecule step-by-step $P(M_i | M_{i+1}, …, M_1, z)$, attaching motif after motif. Each generation step is designed identically and consists of multiple modules, which are repeatedly called over the process of the generation. Importantly, such methods can condition on the intermediate state of a molecule, which OS methods cannot.
>
> &nbsp;
>
> - Although we believe that a better terminology would distinguish between i) attaching nodes or fragments to a partial graph and ii) fully generating specific aspects of a molecule, e.g. the set of motifs, we chose to adopt established terminology from [Zhu et al., 2022] to ensure clarity and maintain consistency. However, we agree with the reviewer that this might lead to misunderstandings and misguide the interpretation of the experimental evaluation.
> - To avoid misunderstandings, we removed or toned down *all* claims that rely on MAGNet being an OS model:
>     - We updated L185-186 to remove the classification of MAGNet as OS model
>     - We updated L412-413, such that MAGNet is described to perform on par with motif-based approaches
>
> &nbsp;
>
> - The classification of the methods in Table 1 was intended to help the reader contextualise the results, as the classification is consistent in itself (MAGNet, PS-VAE and DiGress as OS models), and aims to make up for inconsistencies in the literature. We acknowledge that this discussion needs more room in the paper and we describe the categorisation of methods in more detail in App. C of the updated PDF.  However, if the reviewer believes that the classification for this experiment can lead to confusion more than to help the reader, we are happy to remove the classification from this table entirely.
> - Moreover, we do not consider MAGNet being an OS model to be our main contribution, cf. L74-76 as well as L84-92. To us, leveraging scaffolds instead of motifs as well as our experimental evaluation, which was also highlighted by the reviewer, are the most essential aspects of our work.
>
> &nbsp;
>
> We want to thank the reviewer again for raising their concern and encouraging us to discuss the categorisation of methods in more detail. We hope that our updated manuscript as well as our clarifications have addressed the reviewer’s remaining concern satisfactorily.

---

> > ### Comment · Reviewer_Ry7i · 2024-11-25
> > **Thanks**
> >
> > I have read the rebuttal and the revised version. I truly appreciated the great effort put in by the authors to respond to my doubts. All my concerns have been addressed, therefore I am glad to change my initial judgment. I'll be giving full acceptance and I sincerely hope this paper will make it to the conference.
> >
> > Good luck!

---

> ### Author Response · Authors · 2024-11-28
>
> We truly appreciate your thoughtful review and support for our paper's acceptance at ICLR. Your insights, regarding method classification in particular, were valuable, and we are grateful for your constructive feedback.

---

### Official Review · Reviewer_hxc4 · 2024-11-04

**Soundness:** 3
**Presentation:** 3
**Contribution:** 3
**Rating:** 8
**Confidence:** 4

**Summary:**

This paper proposes a new way to address chunk-based molecule generation. Instead of using fully specified molecular subgraphs (motifs) similarly to prior work, the authors instead abstract out motifs to their connectivity skeletons, which allows for a smaller vocabulary to cover a wider range of possible motif realizations. The authors then show a factorization of the generative procedure that first builds the scaffold by assembling these motif skeletons and then gradually fills in the atom features. The approach is verified on standard generation and optimization benchmarks, showing decent performance.

**Strengths:**

(S1): The authors address an important problem of molecule generation, and identify a reasonable gap in the capabilities of prior models that they then try to address. The proposed approach is motivated clearly.



(S2): The paper is generally well-written. The experiments are conducted across various setups that are common/relevant in this domain. While the empirical performance isn't across-the-board amazing, MAGNet seems to be a generally capable model with good performance overall, while improving on top of previous models in some settings, as well as enabling new capabilities (e.g. more ways to condition the generation on partial information).

**Weaknesses:**

(W1): The authors argue the most direct comparison is to other OS models, and focus on that angle. Setting aside the fact that MAGNet is not purely one-shot as the scaffold multiset `S` is decoded sequentially and autoregressively, even if we agree MAGNet is OS, is there any inherent value that comes from being an OS model? One thing that comes to mind would be faster inference, as stepwise generation models can be expensive due to repeatedly encoding the current partial graph. So, is MAGNet more efficient than sequential decoding models, e.g. MoLeR?



(W2): More interesting conditioning settings depicted in Figure 5 could be explained in more detail. The partial molecule induces a partial scaffold multiset `S`; do you then use this partial multiset and continue generating to get a full multiset, then connect the scaffolds while forcing those of the connections that are implied from the conditioning? I assume extending the multiset `S` with further scaffolds cannot directly take into account the fact that some of the scaffold connections (or scaffold instantiations into specific motifs) are already known from the conditioning, because during training the scaffold multiset extension subnetwork assumes only a multiset of generic scaffolds is known. Could this be an issue causing the model to add scaffolds that don't fit well with the partial molecule?

=== Nitpicks ===

Below I list nitpicks (e.g. typos, grammar errors), which did not have a significant impact on my review score, but it would be good to fix those to improve the paper further.

- Line 147: Denoting the join node as `j` could be an index clash given that the scaffolds being considered are denoted as `i` and `j` earlier in the sentence.

- Line 155: "factories" -> "factorize"

- Line 309: missing space

=== Update 26/11 ===

After the author rebuttal I decided to raise my score from 6 (borderline accept) to 8 (accept).

**Questions:**

See the "Weaknesses" section above for specific questions.

If my questions/concerns are addressed, I would consider raising my score.

---

> ### Author Response · Authors · 2024-11-20
>
> &nbsp;
>
> Thank you for the constructive feedback on our submission. In our response, we aim to clarify the remaining questions and concerns of the reviewer.
>
> &nbsp;
>
> > MAGNet is not purely one-shot (W1)
>
> - We agree with the reviewer’s description and classification of our MAGNet being “not purely one-shot” in the classical sense. As prompted also by reviewer Ry7i, we extend the discussion of this aspect in the updated manuscript and would like to refer the reviewer to both the response to reviewer Ry7i and App. C of the updated PDF.
>
> > Is there any inherent value that comes from being an OS model? (W1)
>
> - As the reviewer correctly pointed out, sequential models incur additional computational overhead due to the repeated encoding of partial molecules, generally making them slower than models that do not condition on partial molecules. This difference is evident when comparing generation speeds, which we discuss in more detail in our response to reviewer 8uqV.
> - However, we believe that models that avoid conditioning on partial molecules have another key advantage: their latent representation must fully capture the essential features of the molecule. In contrast, when conditioning on both the partial molecule and the latent representation, the influence of the latent code is not guaranteed. In OS models like MAGNet, the latent code is central to the generation process and dictates all subsequent steps, ensuring faithfulness to the representation, which we, for example, demonstrate through our displacement analysis presented in Fig.7.
> - The hierarchical architecture is essential for MAGNet. Scaffold representations can vary significantly based on their position within the molecule. For example, the Murcko scaffold CC(C)(C)C often appears as CS(=O)(=O)C in a central position, but as CC(F)(F)F in less central contexts. MAGNet requires a complete understanding of the coarse-grained molecular arrangement to assign the correct representations to each scaffold.
>
> > Could [this conditioning scenario] be an issue causing the model to add scaffolds that do not fit well with the partial molecule? (W2)
> - The hierarchical design of MAGNet, which avoids conditioning on partial molecules, offers greater flexibility for tasks like the one presented in Figure 5, where the model needs to condition on disconnected scaffolds. This type of multi-scaffold conditioning is challenging for sequential models, whereas it aligns naturally with MAGNet's factorised approach. Following the reviewer’s feedback, we have extended the description of this experiment in the updated PDF. In general, we share the concern that within our factorisation it might occur that scaffolds are connected such that they do not match with the conditioning fragment. This is likely linked to the complexity of the conditioning scenario.
> - While we have not experienced a mismatch between the partial molecule and generated molecule so far, we recognise that it may arise in other scenarios or datasets. To support the model in such scenarios, we can envision:
>     - Refined Latent Space Navigation: Sampling latent codes in a subspace of the latent space that is in accordance with the conditioning fragments.
>     - Filtering: Incorporating a post-hoc filtering mechanism to eliminate incoherent samples.
> - It would also be possible to incorporate additional architectural changes as listed below, while preserving the reliance on hierarchical decoding with scaffolds as building blocks:
>    - Additive Conditioning: Introducing an auxiliary term, like a scaffold embedding, or constraint in the generation process that explicitly accounts for scaffold compatibility with the partial molecule.
>    - Hierarchical Conditioning: Augmenting the model’s latent space to encode more detailed inter-scaffold relationships, ensuring compatibility even for disconnected scaffolds.
>    - Correction Step: Allowing the model to iterate over its prediction to ensure consistency between the condition and its final prediction.
>
> > Nitpicks
> - We sincerely thank the reviewer for carefully pointing out the typos in our manuscript. We have corrected these issues and believe that it improves the clarity and presentation of our work.
>
> &nbsp;
>
> In conclusion, we appreciate the reviewer’s positive evaluation of our work and hope to have addressed the concerns raised to the reviewer’s satisfaction.

---

> > ### Comment · Reviewer_hxc4 · 2024-11-25
> >
> > Thank you for your response, and for extending the discussion on various aspects I mentioned.
> >
> > As for generation speed, I took a look at the numbers you shared in the reply to Reviewer 8uqV. It does seem like your method is around 2x faster than MoLeR (which is one of the faster autoregressive methods). I noticed the timings are per 10 molecules, and I wonder how batching affects this. Often (e.g. during optimization), one wants to simultaneously decode hundreds of molecules, allowing for batching and parallelization. Does the speed comparison change when decoding 100+ molecules at a time?

---

> > > ### Author Response · Authors · 2024-11-25
> > >
> > > &nbsp;
> > >
> > > We thank the reviewer for engaging in a discussion with us. Moreover, we are happy to learn that our response addressed the reviewer’s mentioned aspects satisfactorily. Below we provide a detailed answer to the reviewer’s question regarding MAGNet’s scaling behaviour.
> > >
> > > &nbsp;
> > >
> > > > MAGNet is around 2x faster than MoLeR
> > > - We want to highlight that this setting, as correctly pointed out by the reviewer, is related to a small-scale experiment where we aimed to match generation speed with the underlying generation process (OS or sequential).
> > > - Note that all considered models are in the same magnitude of generation time. DiGress, for example, as a diffusion model, is much slower, taking around 30 seconds per 10 molecules.
> > > - In our response to Reviewer 8uqV, we already indicated that these numbers are conflated by the underlying DL framework, general implementation, and/or specific optimisation procedures.
> > >
> > > > Does the speed comparison change when decoding 100+ molecules at a time?
> > > - Yes and no. In a large-scale setting, with large batch size and many thousands of molecules, MAGNet will benefit from larger batch sizes. MoLeR, on the other side, additionally benefits from (model-independent) optimisations specifically aimed at large-scale and parallel execution, leading to faster generation times.
> > > - Note that such optimisations (e.g. static graph execution, multi-GPU inference, advanced batching) are not yet implemented for MAGNet as we considered them subordinate compared to our main contribution of addressing the limited diversity and expressivity of molecular generative models.
> > > - However, during this discussion round, we got the impression that generation speed and, therefore, optimisations targeted at that, are generally relevant for the community. We are confident that we can provide support for fast large-scale inference as part of the final code release for MAGNet. We expect that we will likely match or exceed MoLeR’s generation speed also in the mentioned large-scale setting.
> > >
> > > &nbsp;
> > >
> > > We thank the reviewer for asking this relevant question, giving us the opportunity to clarify the difference between a large-scale setting and the conducted small-scale experiment. We hope that this answers the question to the reviewer’s satisfaction and remain available for further discussion.

---

> > > > ### Comment · Reviewer_hxc4 · 2024-11-26
> > > >
> > > > I see, thank you. Indeed, it would be great for the released code to allow fast inference. In my view, there are several older works that, despite being well-cited, never got used that much by practitioners in the drug discovery community, as they didn't make use of parallelism/batching (and so were too slow to use in practice).
> > > >
> > > > By the way, I also found the new results comparing to MoLeR with a very large vocabulary interesting. It seems believable that this model would indeed struggle to make use of a very large vocabulary efficiently.
> > > >
> > > > I raised my score to a full accept to reflect that most of my concerns were addressed.

---

> > > > > ### Author Response · Authors · 2024-11-28
> > > > >
> > > > > Thank you for your comprehensive review and recommendation for acceptance at ICLR. Your guidance on MAGNet's implementation provides a clear deliverable for the final code release, and we deeply value the time and expertise you invested in evaluating our paper.

---

### Author Response · Authors · 2024-11-20
**General Response to all Reviewers**

&nbsp;

We sincerely thank all reviewers for their valuable feedback, questions, and suggestions. We are particularly encouraged by the positive remarks regarding the novelty of our approach, the critical evaluation of motif-based methods, and the thoroughness of our experimental analysis.

&nbsp;

While we provide detailed responses to individual comments, we would like to highlight the key ways in which the feedback has improved our manuscript:
- As encouraged by reviewers Ry7i and hxc6, we have extended the discussion around the generation procedure terminology (OS vs. Sequential). We believe that the updated PDF better contextualises the classification of individual models, while still guiding the reader through the experimental results.
- Based on the feedback from reviewer hxc6, we have improved the description and explanation of the conditional generation experiments.
- The feedback by reviewers 8uqV and w2Yn prompted us to evaluate MoLeR with a significantly larger vocabulary (2k motifs), strengthening the discussion around the expressivity of our vocabulary and fragmentation:
    - Throughout the experimental section, all methods with a variable vocabulary were trained with the same number of motifs in their vocabulary, extracted from the same dataset, to allow for a fair comparison.
    - MiCaM is one example of a model with a very large vocabulary, >15,000 motifs. Besides issues during the training because of the large vocabulary (reference), we observe throughout our analysis that this does not aid the model in the decoding of rare scaffolds.
    - As an additional baseline for our experiment, we train a MoLeR model with a vocabulary of size 2,000, which is almost 6x the size of the MAGNet vocabulary. In App. D.1 of the updated PDF, we extend our analysis on the sampling and reconstruction of scaffolds. Interestingly, we find that an increased vocabulary can help with the sampling of scaffolds, in particular uncommon ones. However, we can also observe that even with a significantly larger vocabulary, MoLeR is not able to utilise this vocabulary to faithfully reconstruct uncommon scaffolds. We conclude that during unconditional generation, the model samples such scaffolds by chance, but cannot make use of them when it is explicitly given by the latent representation, as it is required when using the latent space for downstream tasks and procedures.

&nbsp;

In summary, we greatly appreciate the constructive feedback we received. The reviewer’s  suggestions have strengthened the claims of our work and improved the overall quality of the manuscript.

---

### Meta-Review · Area_Chair_Zd3Q · 2024-12-18

**Metareview:**

Among the reviewers, there was broad agreement that this work
addresses important and relevant problems for the task of molecule generation. Particularly, the proposed use of scaffolds instead of the (usually used) motifs was considered as an interesting and innovative contribution.

There were also some critical remarks concerning the technical description of related approaches, about potential limitation by only considering fragments (instead of synthesis), and about several details in the experimental evaluation and benchmarking process.

 After the rebuttal and discussion phase, however, I had the impression that most of these concerns could be addressed (at least partially), and I think that finally the strengths outweigh the weaknesses. Therefore I recommend acceptance of this paper

**Additional Comments On Reviewer Discussion:**

Many points of criticism raised by the reviewers could be addressed in the rebuttal.

---

### Decision · Program_Chairs · 2025-01-22

Accept (Spotlight)